# On the Coordination of Value-Maximizing Bidders

**Yanru Guan** [1]  **Jiahao Zhang** [2]  **Zhe Feng** [3]  **Tao Lin** [4][5]

## Abstract

While the auto-bidding literature predominantly considers independent bidding, we investigate the coordination problem among multiple auto-bidders in online advertising platforms. Two motivating scenarios are: collaborative bidding among multiple bidders managed by a third-party bidding agent, and strategic bid selection for multiple ad campaigns managed by a single advertiser. We formalize this coordination problem as a theoretical model and investigate the coordination mechanism where only the highest-value bidder competes with outside bidders, while other coordinated bidders refrain from competing. We demonstrate that such a coordination mechanism dominates independent bidding, improving both Return-on-Spend (RoS) compliance and the total value accrued for the participating auto-bidders or ad campaigns, for a broad class of auto-bidding algorithms. Additionally, our simulations on synthetic and real-world datasets support the theoretical result that coordination outperforms independent bidding. These findings highlight both the theoretical potential and the practical robustness of coordinated auto-bidding in online auctions.

## 1. Introduction

Online advertising is a prominent approach to monetize search engines, news feeds, social media, and e-commerce. Among them, real-time auctions have been widely used to connect advertisers and users efficiently. In modern online advertising platforms, advertisers endeavor to maximize their campaign effectiveness, quantified by conversions or other pertinent metrics, through precise bid management that considers targeted Return-on-Spend (RoS) (Aggarwal et al., 2019), which is also well-known as auto-bidding problem. To this end, a wide spectrum of bidding methodologies has been formulated, leveraging concepts from optimization theory, online learning paradigms, and game-theoretic principles (Zhao et al., 2018; Aggarwal et al., 2019; Lee et al., 2013; Babaioff et al., 2021; Golrezaei et al., 2021; Deng et al., 2021; Balseiro et al., 2021; Gao et al., 2022; Stram et al., 2024).

Despite the wealth of sophisticated research in online advertising, existing literature predominantly concentrates on optimizing the value or utility from the perspective of *individual* bidders. This focus, while valuable, often overlooks a crucial aspect of real-world advertising dynamics: the possibility of *coordinated* bidding strategies. In practice, it is common for multiple bidders to form coalitions, perhaps by authorizing a third-party advertising platform to manage their bids collectively. Alternatively, a single large advertiser, such as major e-commerce players like Amazon, Temu, or Shein, might control numerous distinct advertising campaigns. These entities often have a portfolio of advertisements they are willing to display to users, necessitating a coordinated approach across their various campaigns to maximize overall effectiveness, rather than optimizing each campaign in isolation. Consequently, a deeper understanding of how auto-bidding algorithms can be coordinated across multiple allied bidders or campaigns, and what are the benefits of coordination over independent bidding, is becoming increasingly critical. Such insights might be useful for industry practices and unlock further efficiencies in online advertising markets.

In this work, we investigate the coordination problem among multiple auto-bidding algorithms in repeated auctions. We formulate a theoretical model in which $N$ auto-bidders participate in $T$ rounds of second-price auctions. The values of these bidders are independent and identically distributed (i.i.d.), motivated by the practice that only similar bidders are chosen to compete in ad auctions. These bidders may form a coalition to determine their bids jointly, competing against bidders outside the coalition. Each auto-bidder is modeled as a value-maximizing agent subject to an RoS constraint (i.e., non-negative overall utility). We consider a simple form of coordination: only the bidder with the

---

[1]School of Electronics Engineering and Computer Science, Peking University, Beijing, China [2]School of Computer Science, Carnegie Mellon University, Pittsburgh, PA, USA [3]Google DeepMind, Mountain View, CA, USA [4]Microsoft Research, Cambridge, MA, USA [5]School of Data Science, The Chinese University of Hong Kong, Shenzhen, Guangdong, China. Correspondence to: Tao Lin <lintao@cuhk.edu.cn>.

*Proceedings of the 43rd International Conference on Machine Learning*, Seoul, South Korea. PMLR 306, 2026. Copyright 2026 by the author(s).

highest value inside the coalition submits a positive bid to compete against the outside bidders, while the other $N-1$ coalition members refrain from competing. In contrast, in the independent bidding scenario, all the $N$ bidders submit positive bids to compete against each other and the outside bidders. We aim to compare the coalition bidders' welfare in the coordinated and independent bidding scenarios.

The main contributions of this paper are as follows. Theoretically, we prove that the highest-value-bidder-compete coordination mechanism mentioned above, although straightforward, enjoys a significant advantage over independent bidding: *By coordination, bidders in the coalition acquire higher values, as well as achieve lower RoS constraint violations, compared to independent bidding, for a broad class of auto-bidding algorithms.*

In particular, we identify a condition on the distributions of bidders' values and outside bids – Assumption 3.1 – under which each coalition bidder's total utility in $T$ periods is higher, hence the bidder's RoS constraint is less violated, under coordination than under independent bidding (Theorem 3.1). This conclusion holds for any auto-bidding algorithm that overbids (a common feature for value-maximizing algorithms). Assumption 3.1 is necessary and sufficient: when it does not hold, there exists an overbidding auto-bidding algorithm for which coordination hurts the coalition bidders, reducing their utility (RoS constraint compliance).

We then consider the total value of coalition bidders. We prove that, as long as bidders use *mirror-descent* algorithms – a broad class of state-of-the-art auto-bidding algorithms under RoS constraints – then the total value of the coalition bidders will be improved by coordination, regardless of whether Assumption 3.1 holds (Theorem 4.1). If Assumption 3.1 holds, then such coordinated mirror-descent algorithms not only dominate independent bidding but also achieve the best possible coalition value among *all* possible coordination mechanisms, as $T \to \infty$ (Theorem 4.2).

Beyond the i.i.d. setting, we extend our results to cases where coalition bidders' values are independent but not identically distributed. In this non-i.i.d. setting, we demonstrate that – under slightly stronger assumptions – the same simple coordination algorithm still yields better total value and total RoS violation compared to independent bidding. But in contrast to the i.i.d. case, we cannot guarantee that this algorithm improves the performance for every individual bidder. Designing a coordination mechanism that achieves superior individual results in the non-i.i.d. setting remains an interesting open problem.

Finally, Section 6 demonstrates, by experiments on synthetic and real-world datasets, that our coordinated mechanism consistently outperforms independent bidding, validating our theoretical results under realistic market conditions.

## 1.1. Related Work

Bidder coordination, often referred to as cartels or collusion in economics, has been extensively studied in traditional auction theory (Robinson, 1985; Hendricks & Porter, 1989; Marshall & Marx, 2007). Prior research has studied collusive behavior in first-price auctions (Lopomo et al., 2011; Pesendorfer, 2000), second-price auctions (Mailath & Zemsky, 1991; Graham & Marshall, 1987), collusion-proof mechanism design (Che & Kim, 2009), auction design under collusion (Pavlov, 2008), and collusion detection (Chotibhongs & Arditi, 2012). While these studies typically consider bidders forming collusions *autonomously* and reaching equilibrium outcomes in static auction settings (Bergemann & Morris, 2016; Fu & Lin, 2025), we instead introduce a *central planner* that coordinates bidders to improve their total welfare in dynamic auction environments.

Recently, coordination behavior has also been observed in online advertising auctions (Decarolis et al., 2023), sparking growing interest in understanding and analyzing such phenomena. Decarolis et al. (2020) studied the impact of coordination on the revenue and efficiency of GSP and VCG auctions. Romano et al. (2022) investigated the computational challenges faced by a media agency coordinating bidders under those mechanisms. Chen et al. (2023) analyzed coordinated online bidding in repeated second-price auctions with budget constraints. While those works focus on coordinating *utility maximizers*, we study *value maximizers*. To the best of our knowledge, this is the first work to investigate coordination among value maximizers, which exhibit different properties from utility maximizers.

Our work is also related to the large literature on auto-bidding. For example, Paes Leme et al. (2024) studied the dynamics of systems with multiple auto-bidders and revealed complex behaviors such as bi-stability, periodic orbits, and quasi-periodicity, while Aggarwal et al. (2025) analyzed optimal bidding strategies for advertisers across multiple platforms. For a comprehensive overview of the literature, we refer readers to the survey by Aggarwal et al. (2024). Our study complements the predominant individual-optimization perspective of the auto-bidding literature.

## 2. Model: Coordinated Auto-Bidding

**Repeated Second-Price Auctions with Coalition.** We consider the scenario where $N$ *auto-bidders* (or *bidders*) participate in $T$ rounds of repeated second-price auctions. In each round $t \in [T] = \{1, \ldots, T\}$, one item (e.g., ad slot) is for sale, and each bidder $i \in [N] = \{1, \ldots, N\}$ draws a value $v_{i,t} \in [0, B]$ for the item i.i.d. from a continuous distribution $F$ with full support on $[0, B]$. Let $\boldsymbol{v}_t = (v_{1,t}, \ldots, v_{N,t})$ be the vector of values of all bidders at round $t$. Let $b_{i,t}$ denote bidder $i$'s bid at round $t$. The

$N$ bidders may form a coalition. Our model captures two possible types of coalitions in ad auctions:

1. $N$ advertisers, each with an ad, instruct their auto-bidding algorithms to coordinate with each other.

2. One advertiser owns $N$ ads. Instead of using $N$ auto-bidding algorithms to bid for the $N$ ads independently, the advertiser instructs the algorithms to coordinate.

In either case, usually in practice, only similar advertisers/ads are selected to compete for an ad slot, which justifies the i.i.d. value assumption. In addition, we will show in Section 5 that our results extend to non-i.i.d. value settings.

There is a competing bid $d_t^O \in [0, B]$ from outside of the coalition, assumed to be i.i.d. sampled from another distribution $D$ every round. Our model also captures second-price auctions with reservation price, because $d_t^O$ can be the maximum of the reservation price and the outside competing bid. We denote the competing bid faced by bidder $i$ at round $t$ as

$$d_{i,t} = \max \left\{ d_t^O, \max_{j \in [N] \setminus \{i\}} b_{j,t} \right\}. \tag{1}$$

Let $x_{i,t} := \mathbb{I}\{b_{i,t} \geq d_{i,t}\} \in \{0, 1\}$ indicate whether bidder $i$ wins the item (ignoring tie-breaking for now), so bidder $i$'s payment is $p_{i,t} := x_{i,t} d_{i,t}$ and utility is $u_{i,t} := x_{i,t} v_{i,t} - p_{i,t} = x_{i,t}(v_{i,t} - d_{i,t})$ at round $t$.

**Value Maximization under RoS Constraint.** Each auto-bidder aims to maximize its total value while adhering to the Return on Spend (RoS) constraint. The formal definition of the optimization problem for bidder $i$ is as follows:

$$\max_{\{b_{i,t}\}_{t=1}^T} \sum_{t=1}^T v_{i,t} \cdot x_{i,t} \qquad \text{(bidder } i\text{'s value)}$$

$$\text{s.t.} \quad \sum_{t=1}^T \Big( \underbrace{v_{i,t} \cdot x_{i,t} - p_{i,t}}_{u_{i,t}} \Big) \geq 0 \quad \text{(RoS constraint)}.$$

The RoS constraint ensures that for every dollar spent, at least one dollar of value is generated; it is equivalent to requiring the bidder's total utility to be non-negative.[1]

**Independent and Coordinated Auto-Bidding.** Each auto-bidder runs an algorithm $A$ that learns to bid from history, in order to maximize the bidder's total value subject to RoS constraint. Formally, at each round $t$, bidder $i$'s algorithm $A(\cdot)$ takes the historical information $H_{i,t}$ available to bidder $i$ – for example, the past values, bids, and allocations

---

[1] In fact, the general RoS constraint requires $\sum_{t=1}^T v_{i,t} \cdot x_{i,t} - \tau \cdot \sum_{t=1}^T p_{i,t} \geq 0$, i.e., at least $\tau$ dollar of value will be generated for every dollar spent. However, it is without loss of generality to assume $\tau = 1$ in the above optimization problem because we can rescale the bidders' values by dividing by $\tau$.

of the bidder: $H_{i,t} = \{v_{i,t'}, b_{i,t'}, x_{i,t'}\}_{t'=1}^{t-1} \cup \{v_{i,t}\}$ – as input and outputs the bid $b_{i,t}$ for the current round. We refer to the scenario where the auto-bidding algorithms are run independently as *independent bidding*, formalized below:

---
**Algorithm 1** Independent Bidding
---
**for** $t = 1, 2, \ldots, T$ **do**
  Each bidder $i \in [N]$ observes their value $v_{i,t} \sim F$ and bids $b_{i,t} = A(H_{i,t})$.
  Each bidder $i \in [N]$ obtains allocation $x_{i,t}$ and pays $p_{i,t}$.
**end for**

---

Next, we define a *coordinated bidding* scenario where, at each round, only the highest-value bidder in the coalition places a positive bid while all other bidders bid zero. We assume that ties are resolved naturally.[2] This is formalized in Algorithm 2.

---
**Algorithm 2** Coordinated Bidding
---
**for** $t = 1, 2, \ldots, T$ **do**
  Each bidder $i \in [N]$ observes their value $v_{i,t} \sim F$.
  Let $i^* = \arg\max_{i \in [N]} v_{i,t}$.
  Bidder $i^*$ bids $b_{i^*,t} = A(H_{i^*,t})$.
  Bidder $i \neq i^*$ bids $b_{i,t} = 0$.
  Each bidder $i \in [N]$ obtains allocation $x_{i,t}$ and pays $p_{i,t}$.
**end for**

---

## 3. Coordination Reduces RoS Violation

In this section, we analyze the extent to which the RoS constraint of every auto-bidder is maintained/violated. We will identify the necessary and sufficient condition under which coordinated bidding results in a smaller RoS violation compared to independent bidding (equivalently, every bidder in the coalition obtains a higher utility through coordination), for any auto-bidding algorithm.

Formally, let $U_{i,T}^{C,A}, U_{i,T}^{I,A}$ be the total utility of bidder $i$ during the $T$ rounds using auto-bidding algorithm $A$ under coordinated and independent bidding scenarios, respectively:

$$U_{i,T}^{C,A} = \sum_{t=1}^T u_{i,t}^{C,A} \qquad U_{i,T}^{I,A} = \sum_{t=1}^T u_{i,t}^{I,A}.$$

RoS constraint requires $U_{i,T}^{C \text{ or } I, A} \geq 0$. If $U_{i,T}^{C \text{ or } I, A} \geq -c$, we say the RoS constraint for bidder $i$ is violated by $c$.

We allow the bidders to use *any* auto-bidding algorithm that *(weakly) overbids*: namely, $A(\cdot)$ can be any function that outputs bid $b_{i,t} = A(H_{i,t}) \geq v_{i,t}$. Recall that we consider

---

[2] Ties happen with probability 0 given a continuous $F$.

value maximization in second-price auctions. If bidders underbid ($b_{i,t} < v_{i,t}$), they will always obtain non-negative utility and satisfy the RoS constraint, but not necessarily maximize their values. So we assume overbidding.

We introduce a condition/assumption on the value and outside bid distributions, $F$ and $D$. Let $(x)_+ = \max\{x, 0\}$ denote the positive part of a number. Let $v_{(N)}, v_{(N-1)}$ be the largest and second-largest values among $N$ i.i.d. samples from $F$.

**Assumption 3.1.** In expectation, the advantage of the second-largest value $v_{(N-1)}$ over the outside bid $d^O$ is larger than the advantage of $d^O$ over the largest value $v_{(N)}$:

$$\Delta := \mathbb{E}_{F,D}\big[(v_{(N-1)} - d^O)_+ - (d^O - v_{(N)})_+\big] \geq 0.$$

Intuitively, Assumption 3.1 says that the largest two values among the coalition bidders are competitive enough against the outside bid. This assumption is satisfied when the number $N$ of coalition bidders is large enough:

**Observation 3.1.** *For any full-support distributions $F$ and $D$, Assumption 3.1 is satisfied when $N$ is large enough.*

Assumption 3.1 is also satisfied by some natural distributions $F$ and $D$ with small $N$:

**Example 3.1.** The following examples satisfy Assump. 3.1:

- $N = 4, F = U[0, 1], D = U[0, 1]$, with $\Delta = 1/6$.
- $N = 3, F = U[0, 1], D = \text{Beta}(3, 2)$, with $\Delta = 1/40$.

The main result of this section is the following Theorem 3.1, which shows that Assumption 3.1 is the necessary and sufficient condition for coordination to reduce RoS constraint violation compared to independent bidding. When $\Delta \geq 0$, coordination improves the utility of every bidder in the coalition, for any auto-bidding algorithm that weakly overbids. When $\Delta < 0$, this guarantee fails: coordination can be strictly worse than independent bidding for some overbidding algorithm.

**Theorem 3.1.** *If Assumption 3.1 holds (i.e., $\Delta \geq 0$), then for any (weakly) overbidding auto-bidding algorithm $A$ and every bidder $i \in [N]$,*

$$\mathbb{E}\Big[U_{i,T}^{C,A} - U_{i,T}^{I,A}\Big] \geq \frac{T\Delta}{N} \geq 0.$$

*If Assumption 3.1 fails (i.e., $\Delta < 0$), then there exists an overbidding algorithm $A$ such that, for every bidder $i \in [N]$, for sufficiently large $T$,*

$$\mathbb{E}\Big[U_{i,T}^{C,A} - U_{i,T}^{I,A}\Big] < 0.$$

**Remark 3.1.** We can also obtain high-probability results from Theorem 3.1. Because values and outside bids are

bounded by $B$, we can use Azuma's inequality to prove that: when $\Delta > 0$, for any overbidding algorithm $A$,

$$U_{i,T}^{C,A} \geq U_{i,T}^{I,A} + \frac{T\Delta}{2N}$$

holds with probability at least $1 - \exp\big(-\frac{T\Delta^2}{32B^2N^2}\big)$.

### 3.1. Proof of Theorem 3.1

The proof of the second (negative) direction is in Appendix A.2. Below we prove the first (positive) direction.

Let $U_{i,T}^{\text{Truth}} = \sum_{t=1}^{T} u_{i,t}^{\text{Truth}}$ be the total utility of bidder $i$ when all bidders in the coalition bid their values truthfully: $b_{i,t}^{\text{Truth}} = v_{i,t}$. We will prove Theorem 3.1 by proving $\mathbb{E}[U_{i,T}^{C,A}] \geq \mathbb{E}[U_{i,T}^{\text{Truth}}] + \frac{\Delta T}{N}$ and $U_{i,T}^{\text{Truth}} \geq U_{i,T}^{I,A}$. The second inequality is formalized below:

**Lemma 3.1.** *For any sequence of values and outside bids $(\boldsymbol{v}_t, d_t^O)_{t=1}^T$, $U_{i,T}^{\text{Truth}} \geq U_{i,T}^{I,A}$.*

*Proof.* This lemma directly follows from the dominant-strategy-incentive-compatibility of second-price auction, which ensures that each bidder's truthful-bidding utility $u_{i,t}^{\text{Truth}}$ is weakly larger than the non-truthful bidding utility $u_{i,t}^{I,A}$ at every round. □

We then prove $\mathbb{E}[U_{i,T}^{C,A}] \geq \mathbb{E}[U_{i,T}^{\text{Truth}}] + \frac{T\Delta}{N}$:

**Lemma 3.2.** *Under Assumption 3.1, for any auto-bidding algorithm that overbids, for every bidder $i \in [N]$,*

$$\mathbb{E}\Big[U_{i,T}^{C,A} - U_{i,T}^{\text{Truth}}\Big] = \sum_{t=1}^{T}\Big(u_{i,t}^{C,A} - u_{i,t}^{\text{Truth}}\Big) \geq \frac{T\Delta}{N}.$$

*Proof.* See Appendix A.3. □

Lemmas 3.1 and 3.2 together prove Theorem 3.1.

## 4. Coordination Increases Bidders' Value

In this section, we show that coordination can increase the total value of auto-bidders, compared to independent bidding. This holds for a broad class of auto-bidding algorithms and without any distributional assumptions.

In particular, we consider any *mirror-descent-based* auto-bidding algorithm. They are state-of-the-art algorithms for value maximization under RoS constraint, with near-optimal $O(\sqrt{T})$ regret guarantees (Balseiro et al., 2023; Feng et al., 2023). The algorithm maintains a parameter $\lambda_{i,t}$ for each bidder (which corresponds to the Lagrange multiplier of the dual optimization problem), and the bidder overbids by $b_{i,t} = (1 + \frac{1}{\lambda_{i,t}})v_{i,t}$ at each round. After observing a utility feedback, the bidder adjusts the parameter $\lambda_{i,t}$ in the opposite direction of the utility (i.e., positive utility decreases $\lambda_{i,t}$). The full algorithm is given in Algorithm 3.

**Definition 4.1** (Legendre mirror map and Bregman divergence). A $C^1$ function $h : (0, \infty) \to \mathbb{R}$ is a *Legendre mirror map* if: (i) $h$ is strictly convex on $(0, \infty)$; (ii) $h'$ is a bijection from $(0, \infty)$ onto $\mathbb{R}$ (equivalently, $h'$ is strictly increasing with $\lim_{\lambda \downarrow 0} h'(\lambda) = -\infty$ and $\lim_{\lambda \uparrow \infty} h'(\lambda) = +\infty$). The associated Bregman divergence is

$$D_h(\lambda, \mu) := h(\lambda) - h(\mu) - h'(\mu)(\lambda - \mu), \quad \forall \lambda, \mu > 0.$$

---

**Algorithm 3** Mirror-Descent RoS Auto-Bidder (MD-$h$)

---

Initialize multiplier $\lambda_{i,1} = 1$, learning rate $\alpha = 1/\sqrt{T}$.
**for** $t = 1, 2, \cdots, T$ **do**
  Observe value $v_{i,t}$ and set bid $b_{i,t} = (1 + \frac{1}{\lambda_{i,t}})v_{i,t}$.
  Obtain allocation $x_{i,t}$ and pay $p_{i,t}$, compute realized utility $g_{i,t} = v_{i,t} \cdot x_{i,t} - p_{i,t}$.
  Update the multiplier:

$$\lambda_{i,t+1} = \arg\min_{\lambda > 0} \left\{ \alpha \cdot g_{i,t} \cdot \lambda + D_h(\lambda, \lambda_{i,t}) \right\}.$$

**end for**

---

As an example, Feng et al. (2023) study the mirror-descent-based algorithm with the *entropy mirror map* $h(\lambda) = \lambda \log \lambda - \lambda$, in which case the mirror-descent update is

$$\lambda_{i,t+1} = \lambda_{i,t} \exp(-\alpha g_{i,t}). \tag{2}$$

Feng et al. (2023)'s algorithm guarantees that each bidder's total RoS violation in $T$ rounds is at most $O(\sqrt{T} \log T)$ and total value is at most $O(\sqrt{T})$ away from the optimum.

Let

$$V_t^{C,A} = \sum_{i=1}^{N} v_{i,t} x_{i,t}^{C,A} \qquad V_t^{I,A} = \sum_{i=1}^{N} v_{i,t} x_{i,t}^{I,A}$$

be the total values obtained by the $N$ bidders at round $t$ in the coordinated and independent cases, respectively. We show that, for any mirror-descent-based algorithm $A$, coordination always improves the total value of coalition bidders.

**Theorem 4.1.** *Suppose bidders run a mirror-descent algorithm A (Algorithm 3). As $T \to \infty$, the coalition bidders' total value under coordinated bidding is weakly larger than that under independent bidding:*

$$\lim_{T \to \infty} \mathbb{E}\left[ \frac{1}{T} \sum_{t=1}^{T} V_t^{C,A} - \frac{1}{T} \sum_{t=1}^{T} V_t^{I,A} \right] \geq 0.$$

**Proof sketch.** We analyze the mirror-descent dynamics in the derivative space $y_{i,t} := h'(\lambda_{i,t}) \in (-\infty, +\infty)$, where the mirror-descent update becomes $y_{i,t+1} = y_{i,t} - \alpha g_{i,t}$. Let $G_{(N)}(\lambda)$ be the expected utility of a *single* bidder whose value is distributed according to the highest value $v_{(N)} \sim$

$F_{(N)}$ among $N$ bidders, and who bids using multiplier $\lambda > 0$, competing against the outside bid $d^O \sim D$:

$$G_{(N)}(\lambda) := \mathbb{E}\left[ (v_{(N)} - d^O) \cdot \mathbb{I}\left[ (1 + \tfrac{1}{\lambda})v_{(N)} > d^O \right] \right].$$

Let $V_{(N)}(\lambda) := \mathbb{E}\left[ v_{(N)} \cdot \mathbb{I}[(1 + \tfrac{1}{\lambda})v_{(N)} > d^O] \right]$ denote the expected value of this single bidder.

**Lemma 4.1.** $G_{(N)}(\lambda)$ *is strictly increasing in $\lambda > 0$.*

*Proof.* See Lemma B.1. □

Let $\lambda_\star := \inf\{\lambda > 0 : G_{(N)}(\lambda) \geq 0\}$. When $\lambda_\star = 0$, denote $V_{(N)}(0) := \lim_{\lambda \downarrow 0} V_{(N)}(\lambda)$. When $\lambda_\star > 0$, let $y_\star := h'(\lambda_\star)$. The proof proceeds by proving that the multipliers in the coordinated case converge to $\lambda_\star$ eventually and then upper bounding independent bidding through a single-bidder relaxation.

We first characterize the expected update direction under coordination. Let $H_{t-1} := \bigcup_{i=1}^{N} H_{i,t-1}$ be the joint history up to round $t - 1$.

**Claim 4.1.** *Under coordinated bidding, conditioning on history $H_{t-1}$, the expected realized utility (which serves as the gradient direction for multiplier update) satisfies*

$$\mathbb{E}\left[ g_{i,t} \mid H_{t-1} \right] = \tfrac{1}{N} G_{(N)}(\lambda_{i,t}).$$

When $\lambda_\star > 0$, since $G_{(N)}(\lambda)$ is increasing, we have $G_{(N)}(\lambda_{i,t}) \gtrless 0$ if and only if $\lambda_{i,t} \gtrless \lambda_\star$, giving expected gradient direction $\mathbb{E}[g_{i,t} \mid H_{t-1}] \gtrless 0$, so the mirror-coordinate update pushes $y_{i,t}$ toward $y_\star$. This property allows us to view $(y_{i,t} - y_\star)^2$ as a potential function, which decreases on average whenever $\lambda_{i,t}$ deviates significantly from $\lambda_\star$. The boundary case $\lambda_\star = 0$ is handled by a one-sided exponential potential. These arguments imply:

**Claim 4.2.** *For any $\delta > 0$, the fraction of rounds $t \leq T$ for which the active bidder $i_t^\star = \arg\max_i v_{i,t}$ is away from the reference multiplier $\lambda_\star$ vanishes in expectation as $T \to \infty$.*

The above claim implies that, asymptotically, the active bidder's multiplier in coordinated bidding converges to $\lambda_\star$. As a result, the coalition bidders' total value converges to the expected value obtained by a single bidder whose value is $v_{(N)} \sim F_{(N)}$ and who bids by multiplier $\lambda_\star$:

**Claim 4.3.** *The coordinated time-average total value converges to the single bidder value*

$$\lim_{T \to \infty} \mathbb{E}\left[ \tfrac{1}{T} \sum_{t=1}^{T} V_t^{C,A} \right] = V_{(N)}(\lambda_\star).$$

We then upper bound the time-average total value under independent bidding.

**Claim 4.4.** *Under independent bidding,*

$$\limsup_{T \to \infty} \mathbb{E}\Big[\tfrac{1}{T} \sum_{t=1}^{T} V_t^{I,A}\Big] \ \leq \ V_{(N)}(\lambda_\star).$$

To see this, view $\bar{b}_t = \max_i b_{i,t}$ as the bid of a *virtual single bidder* with value $v_{(N),t}$; its attained value upper bounds the total value attained by the independent coalition. After clipping each bidder's utility below by $-B$, the utility of the virtual bidder dominates the sum of clipped utilities of the coalition bidders, and the MD updates imply a deterministic $o(T)$ RoS violation bound for the virtual bidder. A Lagrangian envelope for the single-bidder problem then upper bounds its average value by $V_{(N)}(\lambda) + \lambda G_{(N)}(\lambda)$ for every $\lambda > 0$, whose infimum is $V_{(N)}(\lambda_\star)$.

Combining Claim 4.4 with Claim 4.3 proves the theorem. Details are in Appendix B.1. □

Theorem 4.1 does not require any assumption on the distributions of values or outside bid, except for boundedness and continuity. We further show that, if Assumption 3.1 holds, then coordinated mirror descent is not only better than independent mirror descent but also weakly better than *any other coordination mechanism*.

**Definition 4.2** (Coordination Mechanism). A *coordination mechanism* $\mathcal{G}$ is any procedure that, at each round $t$, given all realized values $\boldsymbol{v}_t = (v_{1,t}, \ldots, v_{N,t})$, selects bids $(b_{1,t}, \ldots, b_{N,t})$ for the coalition bidders. The mechanism may be history-dependent, i.e., bids at round $t$ may depend arbitrarily on past observations.

**Theorem 4.2.** *Under Assumption 3.1, as $T \to \infty$, any coordinated mirror-descent algorithm $A$ is optimal in maximizing coalition bidders' total value. Formally, for any coordination mechanism $\mathcal{G}$,*

$$\lim_{T \to \infty} \mathbb{E}\Big[\tfrac{1}{T} \sum_{t=1}^{T} V_t^{C,A} \ - \ \tfrac{1}{T} \sum_{t=1}^{T} V_t^{\mathcal{G}}\Big] \geq 0,$$

*where $V_t^{\mathcal{G}}$ is the total value obtained by the $N$ bidders at round $t$ using mechanism $\mathcal{G}$.*

**Proof sketch.** To prove Theorem 4.2, we first reduce the $N$-bidder coordinated scenario to a single-bidder problem where the single bidder's value is the maximum among the coalition members. Using this reduction, we prove an interesting fact: under Assumption 3.1, the multipliers $\lambda_{i,t}^C$ of coordinated bidders converge to 0 in the limit as $t \to \infty$, meaning that the coordinated bidders will eventually submit arbitrarily high bids. This implies that the coordinated coalition will obtain the highest possible value, among all coordination mechanisms. The full proof is presented in Appendix B.2. □

## 5. Extension to Non-i.i.d. Valuations

This section discusses the extension where bidders have independent but non-identically distributed values. In each round $t \in [T]$, bidders $i \in [N]$ draw values $v_{i,t}$ from different $F_i$. The outside bid $d_t^O$ is still drawn from $D$.

**Total RoS.** In the i.i.d. case, coordination reduces the RoS constraint violation for *every* bidder in the coalition, if and only if Assumption 3.1 holds. In the non-i.i.d. case, we show that under the same assumption, coordination reduces the *total* RoS constraint violation among the coalition bidders, namely, the total utility of the coalition is improved. Recall that $U_{i,T}^{C,A} = \sum_{t=1}^{T} u_{i,t}^{C,A}$ and $U_{i,T}^{I,A} = \sum_{t=1}^{T} u_{i,t}^{I,A}$ are the total utilities of bidder $i$ during $T$ rounds under coordinated and independent bidding scenarios.

**Theorem 5.1** (Coordination reduces total RoS violation, non-i.i.d. case). *Under Assumption 3.1 (i.e., $\Delta \geq 0$), for any overbidding algorithm $A$, the total utility among the $N$ coalition bidders satisfies*

$$\mathbb{E}\Big[\sum_{i=1}^{N} U_{i,T}^{C,A}\Big] \ \geq \ \mathbb{E}\Big[\sum_{i=1}^{N} U_{i,T}^{I,A}\Big] + \Delta T.$$

The proof of this theorem is in Appendix C.1.

Although Theorem 5.1 does not guarantee RoS improvement for every bidder, the total RoS improvement is also desirable in practice because the coalition bidders may redistribute the extra utility among themselves.

**Total value.** In the i.i.d. setting, we showed that coordinated mirror descent algorithms achieve higher total value than independent mirror descent algorithms without any assumption, and higher than any coordination mechanism under Assumption 3.1. The proof, which relied on the fact that each bidder has the highest value among the coalition with probability $1/N$, does not apply to the non-i.i.d. case because bidders now differ in how often they are the highest-value coalition member. We therefore impose a per-bidder condition ensuring that each bidder contributes positively whenever they are selected. Concretely, for bidder $i$, define

$$\Delta_i := \mathbb{E}\big[(v_i - d^O)\, \mathbb{I}\{v_i = \max_{j \in [N]} v_j\}\big].$$

**Assumption 5.1** (Positive per-bidder winning margin). For each bidder $i \in [N]$, $\Delta_i > 0$.

**Theorem 5.2** (Coordination increases total value, non-i.i.d. case). *Under Assumption 5.1, coordinated mirror descent algorithms (Algorithm 3) achieve weakly higher total value than independent mirror descent algorithms and any other coordination mechanism $\mathcal{G}$:*

$$\lim_{T \to \infty} \mathbb{E}\Big[\tfrac{1}{T} \sum_{t=1}^{T} V_t^{C,A} - \tfrac{1}{T} \sum_{t=1}^{T} V_t^{\mathcal{G}}\Big] \geq 0.$$

*Table 1.* Performance summary. Confidence intervals no greater than $\pm 0.0025$ are omitted. All totals are normalized by $T$.

| Figure | Setting | $N$ | $T$ | Total Utility (I) | Total Utility (C) | Total Value (I) | Total Value (C) |
|--------|---------|-----|-----|-------------------|-------------------|-----------------|-----------------|
| 1 | i.i.d. | 2 | 4000 | -0.011 | 0.220 | 0.643 | 0.666 |
| 4a | i.i.d. | 4 | 4000 | -0.077 | 0.302 | 0.774 | 0.800 |
| 4b | i.i.d. | 3 | 4000 | -0.049 | 0.153 | 0.712 | 0.748 |
| 2 | non-i.i.d. | 2 | 10000 | 0.049 | 0.219 | 0.715 | 0.718 |
| 5a | non-i.i.d. | 3 | 20000 | -0.014 | 0.258 | 0.619 | 0.633 |
| 5b | non-i.i.d. | 5 | 20000 | -0.062 | 0.619 | 0.814 | 0.819 |
| 3 | Real data | 4 | 20000 | -0.040 | $0.155 \pm 0.012$ | $0.620 \pm 0.016$ | $0.928 \pm 0.003$ |
| 6 | Real data | 5 | 20000 | -0.065 | $0.172 \pm 0.012$ | $0.608 \pm 0.012$ | 0.958 |

See Appendix C.2 for the proof.

The findings in this section suggest that coordination improves upon independent bidding even in the presence of asymmetry inside the coalition, demonstrating the robustness of our conclusions.

## 6. Experiments

To complement our theoretical analysis, we conduct a series of experiments evaluating the performance of coordinated auto-bidding, on both synthetic data and a public real-world dataset (iPinYou (Zhang et al., 2014; Liao et al., 2014)).

Unless otherwise stated, all metrics are averaged over 100 independent simulations; shaded regions in figures depict 95% confidence intervals computed as $\bar{x} \pm 1.96 \cdot s$, where $\bar{x}$ and $s$ are the sample mean and sample standard deviation divided by $\sqrt{100}$.

### 6.1. Experiment Setup

In all experiments, we run Algorithm 3 with mirror map $h(\lambda) = \lambda \log \lambda - \lambda$ (Feng et al., 2023) (i.e., Equation (2)) and learning rate $\alpha = 1/\sqrt{T}$, where $T$ is the total number of auction rounds. The horizon $T$ may vary across experiments because different settings exhibit different convergence rates. The values of $T$ and $N$ (number of coalition bidders) used in each setting are summarized in Table 1.

#### 6.1.1. SYNTHETIC DATA

**Symmetric setting.** Each coalition bidder $i$ draws values i.i.d. from uniform distribution $U[0, 1]$. The outside bid is sampled from $D = U[0, 0.9]$ (Figure 1), $D = U[0, 1]$ (Figure 4a), or $D = \text{Beta}(3, 2)$ (Figure 4b), representing different market conditions faced by the coalition bidders.

**Asymmetric setting.** To test the coordination mechanism under heterogeneity, we sample each bidder's values from distinct families (Uniform, Beta, or truncated Gaussians with bidder-specific parameters). The outside-bid distribution $D$ varies with the scenario: $U[0.2, 0.8]$ in Figure 2, $\text{Beta}(3, 5)$ in Figure 5a, and $\text{Beta}(2, 8)$ in Figure 5b. Asymmetries arise from both inside the coalition and the outside

market conditions.

Full distributional specifications for the synthetic experiments are provided in the figure captions (Figures 1, 4a, 4b, 2, 5a, 5b). All distributions conform to Assumption 3.1.

#### 6.1.2. REAL-WORLD DATA

We also experimented with the public iPinYou dataset (Zhang et al., 2014; Liao et al., 2014), released for a real-time bidding competition organized by iPinYou. We use the Season 2 test split, which contains 2,521,630 auction records from 5 advertisers. Each record provides the bidding (winning) price and the corresponding advertiser ID.

The real-world data impose three limitations: (i) only the winning price is observed, (ii) bidder identities are unavailable, and (iii) there is no label indicating whether any bidders coordinate. To obtain bidder valuations consistent with the data while abstracting away unobserved identities and strategies, we adopt an empirical i.i.d. model: we aggregate all observed winning prices into a common pool and, in each round, draw each coalition bidder's value independently and uniformly at random with replacement from this pool. This is equivalent to sampling i.i.d. values from the empirical price distribution induced by the dataset. Constructing a reliable non-i.i.d. model would require persistent bidder identities or side information about coordination patterns, which are not available in the data.

We normalize all prices to $[0, 1]$ for numerical stability. The outside bidder's bid is drawn from the same empirical distribution and then scaled by a random factor from $U[1, 2]$ to maintain competitive pressure. We report results for coalitions with $N \in \{4, 5\}$ bidders under this setup.

### 6.2. Results

**Metrics.** Across all experiments, we evaluate two metrics for both INDEPENDENT (Algorithm 1) and COORDINATED (Algorithm 2) algorithms: (a) *utility* (achieved value minus payment), (b) *achieved value*. Higher utility indicates lower RoS violation; in particular, utility $\geq 0$ implies that the RoS constraint is satisfied. Figures report per-bidder trajectories, whereas Table 1 aggregates to coalition-level

totals (sum across bidders). All statistics are averaged over 100 independent runs.

**Plot notation.** In the figures, labels of the form `bidder` $i$ `- I` denote bidder $i$ under the Independent scenario, whereas `bidder` $i$ `- C` denotes the Coordinated scenario. Shaded regions show 95% confidence intervals over 100 simulations.

**Results on synthetic data** In the symmetric settings (Figures 1, 4a, 4b), coordination sharply increases both each bidder's utility (hence reducing RoS violation) and the coalition's total achieved value relative to independent bidding. Although our theory only guarantees value improvement at the coalition (total) level, the experiments further show that *every individual bidder* benefits from coordination by obtaining higher value. We observe that per-bidder curves closely track one another due to the i.i.d. value draws.

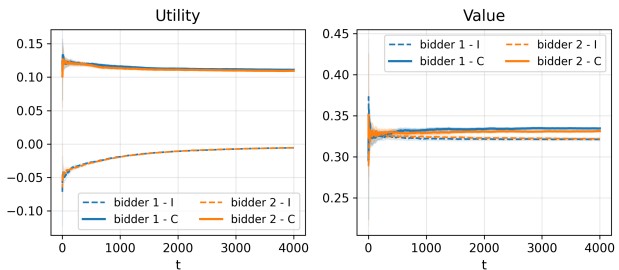

*Figure 1.* Experiments under i.i.d. auto-bidders ($N = 2, F = U[0, 1], D = U[0, 0.9]$).

In the asymmetric settings (Figures 2, 5a, 5b), the aggregate gains largely persist: coordination improves total value on average and typically improves violations. However, heterogeneity induces dispersion in learning dynamics – bidders with different value distributions exhibit different convergence rates, and individual value improvements are not guaranteed for every bidder even though the coalition-level total value increases.

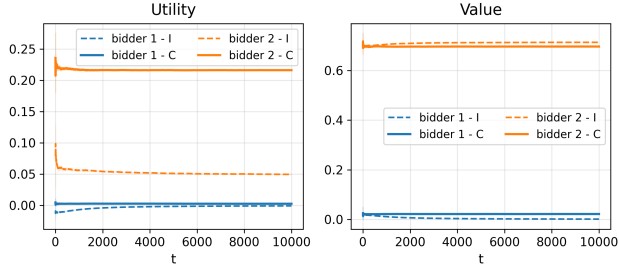

*Figure 2.* Experiments under non-i.i.d. auto-bidders ($N = 2, F_1 = \text{Beta}(2, 5), F_2 = \text{Beta}(5, 2), D = U[0.2, 0.8]$).

**Results on real-world data.** Consistent with the synthetic data experiments, real-world data results (Figures 3, 6) show improvements in both individual RoS violation and achieved value.

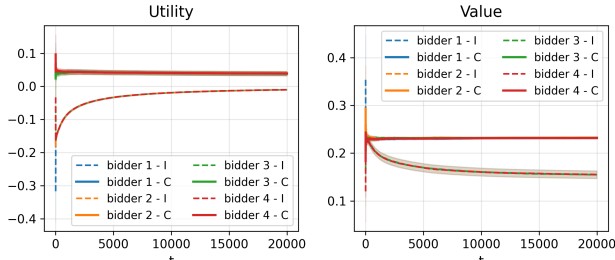

*Figure 3.* Experiments in real-world datasets. ($N = 4$)

## 7. Discussion

In this paper, we studied coordinated auto-bidding mechanisms for value maximization under RoS constraints, to understand how cooperation among learning agents enhances efficiency in repeated auctions. We proposed a simple yet effective coordination protocol, theoretically identified the exact condition under which coordination improves the RoS compliance for every bidder, and proved that coordination unconditionally increases the total value attained by the coalition compared to independent bidding. Our empirical results further validate these findings, showing higher achieved value and lower RoS violation. The demonstrated benefits of coordination over independent bidding might provide useful guidance for industrial practice.

Building on these findings, we outline several directions for future work:

- *Auto-bidding outside bidders:* While we assumed i.i.d. outside bids (a reasonable assumption when outside bidders are uniformly sampled from a population every round), a natural extension is to incorporate *auto-bidding outside bidders*, allowing both coalition members and external participants to adjust bids dynamically.

- *Alternative coordination mechanisms:* While we proved that our highest-value-compete coodination mechanism is asymptotically optimal under Assumption 3.1, it is still interesting to examine whether alternative mechanisms are better when the assumption does not hold. Moreover, if bidders' private values are not accessible, then one may consider other coordination mechanisms, such as highest-bid-compete or mechanisms involving private information elicitation.

- *Other auction formats:* Finally, it would be valuable to extend our analysis to other auction formats, especially non-truthful auctions such as first-price and generalized second-price auctions. Such extensions would help test the robustness of our theoretical insights and clarify how

coordination interacts with strategic bidding incentives in broader market environments.

## Impact Statement

This work is primarily theoretical and aims to advance the understanding of coordinated bidding among value-maximizing agents in auctions. We do not anticipate significant negative societal impacts arising directly from our results. Rather, our results highlight some potential benefits of coordination for bidders and may provide useful guidance for designing or analyzing bidder coordination mechanisms in practice. The broader market-level implications of such coordination – particularly its impact on auctioneer revenue and platform incentives – are not addressed here and represent an interesting direction for future investigation.

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

# A. Omitted Proofs in Section 3

## A.1. Proof of Observation 3.1

Fix $\varepsilon \in (0, B)$. The probability $q_\varepsilon := \Pr_{v \sim F}[v > B - \varepsilon] = 1 - F(B - \varepsilon) > 0$ by full support/continuity of $F$. Let $M_N := \max\{v_1, \ldots, v_N\} = v^{(N)}$ and $S_N := v^{(N-1)}$. The events $\{v_n > B - \varepsilon\}$ are i.i.d. with probability $q_\varepsilon$ and $\sum_{n=1}^{\infty} \Pr(v_n > B - \varepsilon) = \infty$, so by the second Borel–Cantelli lemma, $v_n > B - \varepsilon$ occurs infinitely often for all sufficiently large $n$ with probability one. By monotonicity of the running maxima and of the second maximum, this implies $M_N \geq B - \varepsilon$ and $S_N \geq B - \varepsilon$ for all sufficiently large $N$ with probability one; hence $M_N \to B$ and $S_N \to B$ almost surely.

Define $Z_N := (S_N - d^O)_+ - (d^O - M_N)_+ \in [-B, B]$ and $\Delta_N = \mathbb{E}[Z_N]$. Since $M_N, S_N \to B$ almost surely and $d^O \in [0, B]$, we have $Z_N \to (B - d^O)_+$ almost surely. By the dominated convergence theorem,

$$\lim_{N \to \infty} \Delta_N = \lim_{N \to \infty} \mathbb{E}[Z_N] = \mathbb{E}[\lim_{N \to \infty} Z_N] = \mathbb{E}[(B - d^O)_+] \geq 0,$$

with strict inequality whenever $\Pr(d^O < B) > 0$. Therefore there exists $N_0$ such that $\Delta_N \geq 0$ for all $N \geq N_0$.

## A.2. Proof of the Second (Negative) Result in Theorem 3.1

Suppose Assumption 3.1 does not hold. Let the upper bound on value and outside bid be $B = 1$. For every bidder $i \in [N]$, we construct the following (weakly) overbidding algorithm $A$ that chooses the current bid based on the bidder's past bids:

$$A(H_{i,t}) = \begin{cases} 1, & \text{if there exists some } s < t \text{ with } b_{i,s} = 0, \\ v_{i,t}, & \text{otherwise.} \end{cases}$$

**Claim A.1.** *For every horizon $T \geq 1$, every bidder $i$'s expected total utility difference is*

$$\mathbb{E}\left[U_{i,T}^{C,A} - U_{i,T}^{I,A}\right] = \frac{T}{N} \Delta + \frac{1 - N^{-T}}{N - 1} L. \tag{3}$$

*where $L := \mathbb{E}[(d^O - v^{(N)})_+]$.*

*In particular, if $\Delta < 0$ (i.e., Assumption 3.1 fails with strict negativity), then for all*

$$T > \frac{N}{N - 1} \cdot \frac{L}{-\Delta},$$

*we have $\mathbb{E}[U_{i,T}^{C,A}] < \mathbb{E}[U_{i,T}^{I,A}]$.*

*Proof.* We first note that, under independent bidding, the first case of algorithm $A$ never triggers. Indeed, since $v_{i,t} > 0$ almost surely (by continuity of $F$) and $A$ outputs either $v_{i,t}$ or 1, bidder $i$ never submits a zero bid. Hence the condition $\exists s < t$ with $b_{i,s} = 0$ is never satisfied, so $b_{i,t} = v_{i,t}$ for every $t \in [T]$ and every bidder $i \in [N]$, hence

$$U_{i,T}^{I,A} = U_{i,T}^{\text{Truth}},$$

where $U_{i,T}^{\text{Truth}}$ is bidder $i$'s total utility when all bidders in $[N]$ bid truthfully.

We now analyze the coordinated scenario. In each round $t$, only the highest-value bidder $i_t^*$ follows $A$, while all other bidders bid 0. Fix a bidder $i$ and define

$$\tau_i := \min\{t \geq 1 : i \neq i_t^*\},$$

as the first time when bidder $i$ does not have the highest value, so that bidder $i$ bids 0 for the first time in round $\tau_i$, and the trigger is active from round $\tau_i + 1$ onward. By symmetry of values and independence across rounds,

$$\Pr[\tau_i \geq t] = \Pr[i_1^* = i, \ldots, i_{t-1}^* = i] = \left(\frac{1}{N}\right)^{t-1}.$$

Let $\Delta_{i,t} := u_{i,t}^{C,A} - u_{i,t}^{\text{Truth}}$. Since $U_{i,T}^{C,A} - U_{i,T}^{I,A} = \sum_{t=1}^{T} \Delta_{i,t}$, it suffices to compute $\mathbb{E}[\Delta_{i,t}]$.

- If $\tau_i < t$, bidder $i$ has already bid 0 in the past, so when selected (i.e., having the highest value $v_t^{(N)}$) in round $t$ it bids 1. A direct comparison with truthful bidding shows that on this event

$$\Delta_{i,t} = \underbrace{(v_t^{(N)} - d_t^O)}_{u_{i,t}^{C,A} \text{ when } i \text{ is selected}} - \underbrace{(v_t^{(N)} - \max\{v_t^{(N-1)}, d_t^O\}) \cdot \mathbb{I}[v_t^{(N)} > \max\{v_t^{(N-1)}, d_t^O\}]}_{u_{i,t}^{\text{Truth}} \text{ when } i \text{ has the highest value}}$$

$$= (v_t^{(N)} - d_t^O) - (v_t^{(N)} - \max\{v_t^{(N-1)}, d_t^O\}) \cdot \mathbb{I}[v_t^{(N)} > d_t^O]$$

$$= \mathbb{I}[v_t^{(N)} > d_t^O] \cdot (\max\{v_t^{(N-1)}, d_t^O\} - d_t^O) + \mathbb{I}[v_t^{(N)} \leq d_t^O] \cdot (v_t^{(N)} - d_t^O)$$

$$= (v_t^{(N-1)} - d_t^O)_+ - (d_t^O - v_t^{(N)})_+.$$

By symmetry, bidder $i$ is selected with probability $1/N$, so

$$\mathbb{E}[\Delta_{i,t} \mid \tau_i < t] = \frac{1}{N}\Delta.$$

- If instead $\tau_i \geq t$, bidder $i$ has not yet bid 0, and therefore bids truthfully when selected. In this case the only difference relative to truthful bidding is the reduced payment under coordination, yielding $\Delta_{i,t} = (v_t^{(N-1)} - d_t^O)_+$,

$$\mathbb{E}[\Delta_{i,t} \mid \tau_i \geq t] = \frac{1}{N}\mathbb{E}[(v^{(N-1)} - d^O)_+] = \frac{\Delta + L}{N}.$$

Combining the two cases,

$$\mathbb{E}[\Delta_{i,t}] = \Pr[\tau_i < t]\frac{\Delta}{N} + \Pr[\tau_i \geq t]\frac{\Delta + L}{N} = \frac{\Delta}{N} + \frac{L}{N^t}.$$

Summing over $t = 1, \ldots, T$ gives

$$\mathbb{E}[U_{i,T}^{C,A} - U_{i,T}^{I,A}] = \sum_{t=1}^{T} \mathbb{E}[\Delta_{i,t}] = \frac{T}{N}\Delta + \frac{1 - N^{-T}}{N - 1}L,$$

which proves Equation (3). If $\Delta < 0$, the final inequality follows immediately for $T$ exceeding the stated threshold. $\square$

### A.3. Proof of Lemma 3.2

For any sequence of values and outside bids $(\boldsymbol{v}_t, d_t^O)_{t=1}^T$, consider the difference between bidder $i$'s coordinated auto-bidding utility and truthful bidding utility at round $t$:

$$\Delta_{i,t} = u_{i,t}^{C,A} - u_{i,t}^{\text{Truth}}$$

$$= (v_{i,t} - d_t^O) \cdot \mathbb{I}[b_{i,t}^{C,A} \geq d_t^O] \cdot \mathbb{I}[v_{i,t} = \max_{j \in [N]} v_{j,t}] - (v_{i,t} - d_{i,t}^{\text{Truth}}) \cdot \mathbb{I}[v_{i,t} \geq d_{i,t}^{\text{Truth}}]$$

where $d_{i,t}^{\text{Truth}} = \max\{d_t^O, \max_{j \neq i} v_{j,t}\}$ is the competing bid for bidder $i$ under truthful bidding. We do a case analysis:

- If $v_{i,t} \neq \max_{j \in [N]} v_{j,t}$, then bidder $i$ loses under both coordinated bidding and truthful bidding, so $u_{i,t}^{C,A} = u_{i,t}^{\text{Truth}} = 0$ and $\Delta_{i,t} = 0$.

- If $v_{i,t} = \max_{j \in [N]} v_{j,t}$, then

$$u_{i,t}^{C,A} = (v_{i,t} - d_t^O) \cdot \mathbb{I}[b_{i,t}^{C,A} \geq d_t^O]$$

$$u_{i,t}^{\text{Truth}} = (v_{i,t} - d_t^{\text{Truth}}) \cdot \mathbb{I}[v_{i,t} \geq d_t^O].$$

Because the auto-bidder is overbidding ($v_{i,t} \leq b_{i,t}^{C,A}$), there are three further cases:

  - $d_t^O < v_{i,t} \leq b_{i,t}^{C,A}$: in this case, $\Delta_{i,t} = d_t^{\text{Truth}} - d_t^O \geq 0$.

- $v_{i,t} \leq b_{i,t}^{C,A} < d_t^O$: in this case, the bidder loses under both scenarios, so $\Delta_{i,t} = 0$.
- $v_{i,t} \leq d_t^O \leq b_{i,t}^{C,A}$: in this case, the bidder loses under truthful bidding but wins under coordinated bidding, so
  $\Delta_{i,t} = u_{i,t}^{C,A} = v_{i,t} - d_t^O \leq 0$.

Then, we condition on any history $H_{t-1}$, randomize over $\boldsymbol{v}_t$ and $d_t^O$, and analyze the expectation of $\Delta_{i,t}$. Let $F_{(N)}$ be the distribution of the largest value $v_{(N)}$ among $N$ samples from $F$.

$$\mathbb{E}_{\boldsymbol{v}_t \sim F, d_t^O \sim D}\big[\Delta_{i,t} \mid H_{t-1}\big]$$
$$= 0 + \Pr[v_{i,t} = \max_{j \in [N]} v_{j,t}] \cdot \mathbb{E}_{v_{i,t} \sim F_{(N)}, d_t^O \sim D}\big[\Delta_{i,t} \mid H_{t-1}\big]$$
$$= \frac{1}{N}\bigg(\Pr_{v_{i,t} \sim F_{(N)}, d_t^O \sim D}\big[v_{i,t} > d_t^O\big] \cdot \mathbb{E}\big[d_t^{\text{Truth}} - d_t^O \mid v_{i,t} \geq d_t^O\big]$$
$$+ \Pr_{v_{i,t} \sim F_{(N)}, d_t^O \sim D}\big[v_{i,t} \leq d_t^O \leq b_{i,t}^{C,A}\big] \cdot \mathbb{E}\big[v_{i,t} - d_t^O \mid v_{i,t} \leq d_t^O \leq b_{i,t}^{C,A}\big]\bigg)$$
$$= \frac{1}{N}\bigg(\mathbb{E}\big[\mathbb{I}[v_{(N)} > d_t^O] \cdot \big(\max\{v_{(N-1)}, d_t^O\} - d_t^O\big)\big] + \mathbb{E}\big[\mathbb{I}[v_{(N)} \leq d_t^O \leq b_{i,t}^{C,A}] \cdot (v_{(N)} - d_t^O)\big]\bigg)$$

Because $\max\{v_{(N-1)}, d_t^O\} - d_t^O > 0$ implies $\mathbb{I}[v_{(N)} > d_t^O] = 1$, the above is equal to

$$= \frac{1}{N}\bigg(\mathbb{E}\big[\max\{v_{(N-1)}, d_t^O\} - d_t^O\big] + \mathbb{E}\big[\mathbb{I}[v_{(N)} \leq d_t^O \leq b_{i,t}^{C,A}] \cdot (v_{(N)} - d_t^O)\big]\bigg)$$
$$\geq \frac{1}{N}\bigg(\mathbb{E}\big[\max\{v_{(N-1)}, d_t^O\} - d_t^O\big] + \mathbb{E}\big[\mathbb{I}[v_{(N)} \leq d_t^O] \cdot (v_{(N)} - d_t^O)\big]\bigg)$$
$$= \frac{1}{N}\bigg(\mathbb{E}\big[(v_{(N-1)} - d_t^O)_+\big] - \mathbb{E}\big[(d_t^O - v_{(N)})_+\big]\bigg).$$

Using Assumption 3.1, we obtain

$$\mathbb{E}_{\boldsymbol{v}_t \sim F, d_t^O \sim D}[\Delta_{i,t} \mid H_{t-1}] \geq \frac{\Delta}{N}.$$

Therefore,

$$\mathbb{E}\big[U_{i,T}^{C,A} - U_{i,T}^{\text{Truth}}\big] = \mathbb{E}\bigg[\sum_{t=1}^{T} \Delta_{i,t}\bigg] \geq T \cdot \frac{\Delta}{N},$$

which proves the lemma.

## B. Omitted Proofs in Section 4

Denote the time-average total values among the $N$ coalition bidders by $V_T^{C,A} := \frac{1}{T} \sum_{t=1}^{T} V_t^{C,A}$ and $V_T^{I,A} := \frac{1}{T} \sum_{t=1}^{T} V_t^{I,A}$ for the coordinated and independent cases, respectively.

### B.1. Proof of Theorem 4.1

Let $v_{(N)} = \max\{v_1, \ldots, v_N\}$ denote the highest value among $N$ i.i.d. samples from $F$. Let $F_{(N)}$ be the distribution of $v_{(N)}$. Let $G_{(N)}(\lambda)$ be the expected utility of a single bidder with value $v_{(N)} \sim F_{(N)}$ and bidding using multiplier $\lambda > 0$, competing against the outside bid $d^O \sim D$:

$$G_{(N)}(\lambda) := \mathbb{E}_{v_{(N)} \sim F_{(N)}, d^O \sim D}\big[(v_{(N)} - d^O) \cdot \mathbb{I}[(1 + \tfrac{1}{\lambda})v_{(N)} > d^O]\big].$$

Define the corresponding single-bidder expected value as

$$V_{(N)}(\lambda) := \mathbb{E}_{v_{(N)} \sim F_{(N)}, d^O \sim D}\big[v_{(N)}\mathbb{I}[(1 + \tfrac{1}{\lambda})v_{(N)} > d^O]\big].$$

Since $V_{(N)}(\lambda)$ is monotone and bounded, the right limit $V_{(N)}(0^+) := \lim_{\lambda \downarrow 0} V_{(N)}(\lambda)$ exists. Define the stable reference multiplier by

$$\lambda_\star := \inf \{\lambda > 0 : G_{(N)}(\lambda) \geq 0\} \in [0, \infty),$$

and denote $V_{(N)}(0) = V_{(N)}(0^+)$ when $\lambda_\star = 0$.

We present the monotonicity property of $G_{(N)}(\lambda)$.

**Lemma B.1** (Monotonicity of $G_{(N)}$). *Assume that the outside-bid distribution $D$ admits a density $f_D$ that is strictly positive on $[0, B]$. Then $G_{(N)}(\lambda)$ is continuous and strictly increasing on $(0, \infty)$. Moreover, $\{\lambda > 0 : G_{(N)}(\lambda) \geq 0\}$ is nonempty. Hence, if $\lambda_\star > 0$, then $G_{(N)}(\lambda_\star) = 0$, while if $\lambda_\star = 0$, then $G_{(N)}(\lambda) > 0$ for every $\lambda > 0$.*

*Proof.* Continuity follows from dominated convergence and the continuity of $D$. Write the single-bidder's expected utility as $G_{(N)}(\lambda) = \mathbb{E}_{v \sim F_{(N)}} G(\lambda; v)$, where $G(\lambda; v)$ is the bidder's expected utility conditioning on having value $v$. Namely,

$$G(\lambda; v) = \mathbb{E}_{d^O \sim D} \left[ (v - d^O) \, \mathbb{I} \left\{ \left(1 + \tfrac{1}{\lambda}\right) v > d^O \right\} \right] = v \, x \left( \left(1 + \tfrac{1}{\lambda}\right) v \right) - p \left( \left(1 + \tfrac{1}{\lambda}\right) v \right),$$

where $x(b)$ is the probability of winning when the bidder bids $b$, and $p(b)$ is the expected payment. Extending $f_D$ by zero outside $[0, B]$, we have

$$x(b) = \int_0^b f_D(z) \, \mathrm{d}z, \qquad p(b) = \int_0^b z f_D(z) \, \mathrm{d}z.$$

Since the second-price auction is truthful, Myerson's payment identity gives

$$p(b) = x(b)b - \int_0^b x(z) \, \mathrm{d}z.$$

Therefore,

$$G(\lambda; v) = vx \left( \left(1 + \tfrac{1}{\lambda}\right) v \right) - \left(1 + \tfrac{1}{\lambda}\right) vx \left( \left(1 + \tfrac{1}{\lambda}\right) v \right) + \int_0^{(1+1/\lambda)v} x(z) \, \mathrm{d}z$$

$$= -\frac{1}{\lambda} vx \left( \left(1 + \tfrac{1}{\lambda}\right) v \right) + \int_0^{(1+1/\lambda)v} x(z) \, \mathrm{d}z.$$

Taking the derivative with respect to $\lambda$,

$$\frac{\partial G(\lambda; v)}{\partial \lambda} = \frac{1}{\lambda^2} vx \left( \left(1 + \tfrac{1}{\lambda}\right) v \right) - \frac{1}{\lambda} vx' \left( \left(1 + \tfrac{1}{\lambda}\right) v \right) \left( -\frac{v}{\lambda^2} \right) + x \left( \left(1 + \tfrac{1}{\lambda}\right) v \right) \left( -\frac{v}{\lambda^2} \right)$$

$$= \frac{v^2}{\lambda^3} x' \left( \left(1 + \tfrac{1}{\lambda}\right) v \right)$$

$$= \frac{v^2}{\lambda^3} f_D \left( \left(1 + \tfrac{1}{\lambda}\right) v \right).$$

Integrating over $v \sim F_{(N)}$,

$$\frac{\mathrm{d}G(\lambda)}{\mathrm{d}\lambda} = \mathbb{E}_{v \sim F_{(N)}} \left[ \frac{v^2}{\lambda^3} f_D \left( \left(1 + \tfrac{1}{\lambda}\right) v \right) \right].$$

Because $F_{(N)}$ has full support on $[0, B]$ and $f_D(z) > 0$ on $[0, B]$, the integrand is positive with positive probability: for example, $v \in (0, B/(1 + 1/\lambda))$ has positive probability, and then $(1 + 1/\lambda)v \in (0, B)$. Hence $\mathrm{d}G(\lambda)/\mathrm{d}\lambda > 0$, so $G$ is strictly increasing.

Finally, because $F_{(N)}$ and $D$ have ful support on $[0, B]$,

$$\lim_{\lambda \to \infty} G(\lambda) = \mathbb{E} \left[ (v_{(N)} - d^O) \mathbb{I}\{v_{(N)} > d^O\} \right] > 0,$$

so $\{\lambda > 0 : G(\lambda) \geq 0\}$ is nonempty. The remaining claims follow from continuity and strict monotonicity. $\square$

For the MD-$h$ update, write $y_{i,t} := h'(\lambda_{i,t}) \in (-\infty, +\infty)$ for the mirror coordinate of bidder $i$'s multiplier. By first-order optimality, the mirror-descent update rule is

$$y_{i,t+1} = y_{i,t} - \alpha g_{i,t}. \tag{4}$$

**Step 1: Coordinated value converges to $V_{(N)}(\lambda_\star)$**

In this step all quantities refer to the coordinated process. Let $H_{t-1}$ denote the joint history before round $t$, and define the drift in mirror coordinates by

$$\phi(y) := \tfrac{1}{N} G_{(N)}\left((h')^{-1}(y)\right), \quad \forall y \in \mathbb{R}.$$

Under coordination, bidder $i$ is active if and only if it is the highest-value bidder in the coalition. Since the values are i.i.d., each bidder is the highest-value bidder with probability $1/N$. So conditioning on the history $H_{t-1}$,

$$\mathbb{E}\big[g_{i,t} \mid H_{t-1}\big] = \tfrac{1}{N} G_{(N)}(\lambda_{i,t}) = \phi(y_{i,t}). \tag{5}$$

Also, in the coordinated process, $g_{i,t} \in [-B, B]$ almost surely.

**Lemma B.2** (Interior occupation bound). *Assume $\lambda_\star > 0$. Let $y_\star := h'(\lambda_\star)$. For every bidder $i \in [N]$ and every $\delta > 0$,*

$$\frac{1}{T} \sum_{t=1}^{T} \mathbb{P}\big(|y_{i,t} - y_\star| \geq \delta\big) \leq \frac{(y_{i,1} - y_\star)^2 + B^2}{2\delta c_\delta} \cdot \frac{1}{\sqrt{T}},$$

*where*

$$c_\delta := \min\{-\phi(y_\star - \delta),\, \phi(y_\star + \delta)\} > 0.$$

*Proof.* Since $\phi$ is strictly increasing and $\phi(y_\star) = 0$, $c_\delta > 0$. For all $y$ with $|y - y_\star| \geq \delta$,

$$(y - y_\star)\phi(y) \geq \delta c_\delta. \tag{6}$$

Let $W_t := (y_{i,t} - y_\star)^2$. By Equation (4),

$$W_{t+1} = W_t - 2\alpha(y_{i,t} - y_\star)g_{i,t} + \alpha^2 g_{i,t}^2.$$

Taking conditional expectation and using Equation (5) and $g_{i,t}^2 \leq B^2$,

$$\mathbb{E}[W_{t+1} \mid H_{t-1}] \leq W_t - 2\alpha(y_{i,t} - y_\star)\phi(y_{i,t}) + \alpha^2 B^2.$$

Summing over $t = 1, \ldots, T$, using $W_{T+1} \geq 0$, and using $\alpha^2 T = 1$, we obtain

$$2\alpha \sum_{t=1}^{T} \mathbb{E}\big[(y_{i,t} - y_\star)\phi(y_{i,t})\big] \leq W_1 + B^2.$$

By Equation (6),

$$\mathbb{E}\big[(y_{i,t} - y_\star)\phi(y_{i,t})\big] \geq \delta c_\delta\, \mathbb{P}\big(|y_{i,t} - y_\star| \geq \delta\big).$$

Dividing by $2\alpha T$ gives the claim. $\qquad\square$

**Lemma B.3** (Boundary occupation bound). *Assume $\lambda_\star = 0$. For every bidder $i \in [N]$ and any $\delta > 0$,*

$$\frac{1}{T} \sum_{t=1}^{T} \mathbb{P}\big(\lambda_{i,t} \geq \delta\big) = O(T^{-1/2}).$$

*Proof.* Since $\lambda_\star = 0$, Lemma B.1 implies $G_{(N)}(\lambda) > 0$ for all $\lambda > 0$. Hence $\phi(y) > 0$ for any $y \in \mathbb{R}$.

Fix $M \in \mathbb{R}$. It suffices to prove

$$\frac{1}{T} \sum_{t=1}^{T} \mathbb{P}\big(y_{i,t} \geq M\big) = O(T^{-1/2}).$$

Let $W_t := \exp(y_{i,t})$. By Equation (4), $W_{t+1} = W_t \exp(-\alpha g_{i,t})$. Using $e^{-x} \leq 1 - x + x^2 e^{|x|}/2$ and $|g_{i,t}| \leq B$, we have

$$\exp(-\alpha g_{i,t}) - 1 \leq -\alpha g_{i,t} + \frac{1}{2}\alpha^2 g_{i,t}^2 e^{\alpha|g_{i,t}|} \leq -\alpha g_{i,t} + \frac{B^2 e^B}{2}\alpha^2$$

Therefore,

$$\mathbb{E}\big[W_{t+1} - W_t \mid H_{t-1}\big] = W_t\mathbb{E}\big[\exp(-\alpha g_{i,t}) - 1 \mid H_{t-1}\big] \leq -\alpha\phi(y_{i,t})W_t + \frac{B^2 e^B}{2}\alpha^2 W_t.$$

On $\{y_{i,t} \geq M\}$, monotonicity gives $\phi(y_{i,t}) \geq \phi(M) > 0$, while outside this event $\phi(y_{i,t}) > 0$. Thus

$$\mathbb{E}\big[W_{t+1} - W_t\big] \leq -\alpha\phi(M)\mathbb{E}\big[W_t\mathbb{I}\{y_{i,t} \geq M\}\big] + \frac{B^2 e^B}{2}\alpha^2\mathbb{E}\big[W_t\big]. \tag{7}$$

Hoeffding's lemma and Equation (5) give

$$\mathbb{E}\big[\exp(-\alpha g_{i,t}) \mid H_{t-1}\big] \leq \exp\left(-\alpha\phi(y_{i,t}) + \frac{\alpha^2 B^2}{2}\right) \leq \exp\left(\frac{\alpha^2 B^2}{2}\right).$$

Therefore, since $\alpha^2 T = 1$,

$$\sup_{t \leq T+1}\mathbb{E}[W_t] \leq \mathbb{E}[W_1]\exp\left(\frac{B^2}{2}\right).$$

Summing (7), using $W_{T+1} \geq 0$, and using $\alpha^2 T = 1$, we get

$$\alpha\phi(M)\sum_{t=1}^{T}\mathbb{E}\big[W_t\mathbb{I}\{y_{i,t} \geq M\}\big] \leq \mathbb{E}[W_1] + \frac{B^2 e^B}{2}\mathbb{E}[W_1]\exp\left(\frac{B^2}{2}\right).$$

Since $W_t \geq e^M$ on $\{y_{i,t} \geq M\}$,

$$\sum_{t=1}^{T}\mathbb{P}\big(y_{i,t} \geq M\big) \leq \frac{e^{-M}}{\alpha\phi(M)}\left(\mathbb{E}[W_1] + \frac{B^2 e^B}{2}E[W_1]\exp\left(\frac{B^2}{2}\right)\right) = O(\alpha^{-1}) = O(\sqrt{T}).$$

Dividing by $T$ gives the desired occupation bound. Finally, taking $M = h'(\delta)$ proves the lemma. □

**Lemma B.4** (Coordinated value limit). *Under coordinated bidding,*

$$\lim_{T\to\infty}\mathbb{E}\big[V_T^{C,A}\big] = V_{(N)}(\lambda_\star).$$

*Proof.* Let $i_t^\star := \arg\max_{i\in[N]} v_{i,t}$ be the highest-value bidder in round $t$. Under coordination,

$$V_t^{C,A} = v_{(N),t} \cdot \mathbb{I}\left\{(1 + \tfrac{1}{\lambda_{i_t^\star,t}})v_{(N),t} > d_t^O\right\}.$$

Hence the expected $T$-round total value is

$$\mathbb{E}\big[V_T^{C,A}\big] = \mathbb{E}\left[\frac{1}{T}\sum_{t=1}^{T}V_{(N)}(\lambda_{i_t^\star,t})\right]. \tag{8}$$

- First suppose $\lambda_\star > 0$. Fix $\varepsilon > 0$. By continuity of $V_{(N)}$ at $\lambda_\star$, choose $\delta > 0$ such that

$$\big|V_{(N)}((h')^{-1}(y)) - V_{(N)}(\lambda_\star)\big| \leq \varepsilon \quad \text{whenever } |y - y_\star| < \delta.$$

Using $0 \leq V_{(N)}(\cdot) \leq B$,

$$\left|\mathbb{E}[V_T^{C,A}] - V_{(N)}(\lambda_\star)\right| \leq \varepsilon + 2B\,\mathbb{E}\left[\frac{1}{T}\sum_{t=1}^{T}\mathbb{I}\{|y_{i_t^\star,t} - y_\star| \geq \delta\}\right].$$

Conditional on $H_{t-1}$, the index $i_t^\star$ is uniform on $[N]$. Thus

$$\mathbb{E}\big[\mathbb{I}\{|y_{i_t^\star,t} - y_\star| \geq \delta\} \mid H_{t-1}\big] = \frac{1}{N}\sum_{i=1}^{N}\mathbb{I}\{|y_{i,t} - y_\star| \geq \delta\}.$$

Lemma B.2 implies that the second term vanishes as $T \to \infty$. Letting $\varepsilon \downarrow 0$ proves the interior case.

- Then suppose $\lambda_\star = 0$. Fix $\varepsilon > 0$. By the definition of $V_{(N)}(0^+)$, choose $\delta > 0$ such that

$$|V_{(N)}(\lambda) - V_{(N)}(0^+)| \le \varepsilon \qquad \forall \lambda \in (0, \delta].$$

Then, by Equation (8),

$$\left| \mathbb{E}\big[V_T^{C,A}\big] - V_{(N)}(0^+) \right| \le \varepsilon + 2B\, \mathbb{E}\left[ \frac{1}{T} \sum_{t=1}^{T} \mathbb{I}\{\lambda_{i_t^\star,t} > \delta\} \right].$$

Again, conditional on $H_{t-1}$,

$$\mathbb{E}\big[\mathbb{I}\{\lambda_{i_t^\star,t} > \delta\} \mid H_{t-1}\big] = \frac{1}{N} \sum_{i=1}^{N} \mathbb{I}\{\lambda_{i,t} > \delta\}.$$

Lemma B.3 implies that the second term vanishes as $T \to \infty$. Since $\varepsilon$ is arbitrary, the boundary case follows. $\qquad\square$

### Step 2: Virtual RoS feasibility under independent bidding

In this step all quantities refer to the independent bidding case, and we sometimes omit the superscript $I, A$. Define

$$\bar{b}_t := \max_{i \in [N]} b_{i,t}, \qquad \bar{V}_t := v_{(N),t}\mathbb{I}\{\bar{b}_t > d_t^O\}, \qquad \bar{g}_t := (v_{(N),t} - d_t^O)\mathbb{I}\{\bar{b}_t > d_t^O\}.$$

We view $\bar{b}_t$ as the bid of a virtual bidder who has value $v_{(N),t}$, competes against the outside bid $d_t^O$ only, and obtains value $\bar{V}_t$ and utility $\bar{g}_t$. Ties have probability zero under the continuity assumptions, and all pointwise statements below are understood almost surely.

For each bidder $i \in [N]$, define the clipped realized utility

$$\widetilde{g}_{i,t} := \max\{g_{i,t}, -B\}.$$

**Lemma B.5** (Virtual utility dominates clipped realized utilities). *For every round $t$, almost surely,*

$$V_t^{I,A} \le \bar{V}_t, \qquad \bar{g}_t \ge \sum_{i=1}^{N} \widetilde{g}_{i,t}.$$

*Proof.* Fix a round $t$. If $\bar{b}_t \le d_t^O$, then no coalition bidder wins against the outside bid. Hence $V_t^{I,A} = 0$, $g_{i,t} = 0$ for all $i$, and $\bar{V}_t = \bar{g}_t = 0$.

Now suppose $\bar{b}_t > d_t^O$. Let $j_t$ be the coalition winner, and let $b_t^{(2)}$ be the second-highest coalition bid. Then

$$V_t^{I,A} = v_{j_t,t} \le v_{(N),t} = \bar{V}_t.$$

Only bidder $j_t$ has nonzero realized utility, and

$$g_{j_t,t} = v_{j_t,t} - \max\{d_t^O, b_t^{(2)}\}.$$

Since $v_{j_t,t} \le v_{(N),t}$ and $\max\{d_t^O, b_t^{(2)}\} \ge d_t^O$,

$$g_{j_t,t} \le v_{(N),t} - d_t^O = \bar{g}_t.$$

Also $\bar{g}_t \ge -B$, because $v_{(N),t}, d_t^O \in [0, B]$. Therefore

$$\sum_{i=1}^{N} \widetilde{g}_{i,t} = \max\{g_{j_t,t}, -B\} \le \bar{g}_t.$$

$\qquad\square$

**Lemma B.6** (Self-bounding of clipped utilities). *For every bidder $i$ and every round $t$,*

$$-B \le \widetilde{g}_{i,t} \le B, \qquad \widetilde{g}_{i,t} \ge -\frac{B}{\lambda_{i,t}},$$

*and in mirror coordinates,*

$$y_{i,t+1} \ge y_{i,t} - \alpha \widetilde{g}_{i,t}.$$

*Proof.* The lower bound $\widetilde{g}_{i,t} \geq -B$ holds by definition. Also $g_{i,t} \leq v_{i,t}x_{i,t} \leq B$, so $\widetilde{g}_{i,t} \leq B$.

If bidder $i$ loses, then $g_{i,t} = 0$, and hence $\widetilde{g}_{i,t} = 0 \geq -B/\lambda_{i,t}$. If bidder $i$ wins, then the competing bid is at most its bid:

$$d_{i,t} \leq b_{i,t} = \left(1 + \tfrac{1}{\lambda_{i,t}}\right) v_{i,t}.$$

Thus

$$g_{i,t} = v_{i,t} - d_{i,t} \geq v_{i,t} - \left(1 + \tfrac{1}{\lambda_{i,t}}\right) v_{i,t} = -\tfrac{v_{i,t}}{\lambda_{i,t}} \geq -\tfrac{B}{\lambda_{i,t}}.$$

Since $\widetilde{g}_{i,t} \geq g_{i,t}$, this proves $\widetilde{g}_{i,t} \geq -\tfrac{B}{\lambda_{i,t}}$.

Recall that the MD update is $y_{i,t+1} = y_{i,t} - \alpha g_{i,t}$. Because $\widetilde{g}_{i,t} \geq g_{i,t}$,

$$y_{i,t+1} = y_{i,t} - \alpha g_{i,t} \geq y_{i,t} - \alpha \widetilde{g}_{i,t}.$$

$\square$

**Lemma B.7** (Deterministic clipped violation bound). *Fix bidder $i$. For any $L > \lambda_{i,1}$,*

$$-\sum_{t=1}^{T} \widetilde{g}_{i,t} \leq \frac{h'(L) - h'(\lambda_{i,1})}{\alpha} + B + \frac{BT}{L}.$$

*Proof.* Define

$$S_t := -\sum_{s=1}^{t} \widetilde{g}_{i,s}, \qquad S_0 := 0.$$

By Lemma B.6,

$$y_{i,t+1} \geq y_{i,t} - \alpha \widetilde{g}_{i,t}.$$

Iterating gives

$$y_{i,t} \geq y_{i,1} + \alpha S_{t-1} \qquad \forall t \geq 1. \tag{6}$$

Let

$$A_L := \frac{h'(L) - h'(\lambda_{i,1})}{\alpha}.$$

If $S_T \leq A_L$, the claim is immediate. Otherwise, define the last time at which the cumulative clipped violation is still below $A_L$:

$$\tau := \max\left\{t \in \{0, 1, \ldots, T\} : S_t \leq A_L\right\}.$$

Then $\tau \leq T - 1$, $S_\tau \leq A_L$, and

$$S_{t-1} > A_L \qquad \forall t \geq \tau + 2.$$

For every $t \geq \tau + 2$, Inequality (6) gives

$$y_{i,t} > y_{i,1} + \alpha A_L = h'(\lambda_{i,1}) + h'(L) - h'(\lambda_{i,1}) = h'(L).$$

Since $h'$ is strictly increasing, $\lambda_{i,t} > L$. Hence, by Lemma B.6,

$$-\widetilde{g}_{i,t} \leq \frac{B}{\lambda_{i,t}} < \frac{B}{L} \qquad \forall t \geq \tau + 2.$$

Using also $-\widetilde{g}_{i,\tau+1} \leq B$, we get

$$S_T = S_\tau + (-\widetilde{g}_{i,\tau+1}) + \sum_{t=\tau+2}^{T} (-\widetilde{g}_{i,t}) \leq A_L + B + \frac{BT}{L}.$$

Since $S_T = -\sum_{t=1}^{T} \widetilde{g}_{i,t}$, the proof is complete. $\square$

**Lemma B.8** (Virtual bidder's RoS violation). *There exists a deterministic sequence $R_T = o(T)$ such that, pathwise,*

$$\sum_{t=1}^{T} \bar{g}_t \geq -N R_T.$$

*Proof.* Because $\lambda_{i,1} = 1$ and $\alpha = 1/\sqrt{T}$, define

$$L_T := (h')^{-1}\left(h'(1) + T^{1/4}\right), \qquad R_T := \frac{h'(L_T) - h'(1)}{\alpha} + B + \frac{BT}{L_T} = T^{3/4} + B + \frac{BT}{L_T}.$$

Since $h'$ is onto and strictly increasing, $L_T \to \infty$ as $T \to \infty$, and hence

$$\frac{R_T}{T} = T^{-1/4} + \frac{B}{T} + \frac{B}{L_T} \to 0.$$

Thus $R_T = o(T)$.

Applying Lemma B.7 with $L = L_T$ gives, for every bidder $i$,

$$\sum_{t=1}^{T} \widetilde{g}_{i,t} \geq -R_T.$$

By Lemma B.5,

$$\sum_{t=1}^{T} \bar{g}_t \geq \sum_{t=1}^{T} \sum_{i=1}^{N} \widetilde{g}_{i,t} = \sum_{i=1}^{N} \sum_{t=1}^{T} \widetilde{g}_{i,t} \geq -NR_T.$$

$\square$

**Step 3: Dual-envelope upper bound on independent value**

For $\lambda > 0$, define

$$\Phi_\lambda(v, b) := \mathbb{E}_{d^O \sim D}\left[v\mathbb{I}\{b > d^O\} + \lambda(v - d^O)\mathbb{I}\{b > d^O\}\right],$$

$$\Psi_\lambda(v) := \sup_{b \geq 0} \Phi_\lambda(v, b), \qquad \mathcal{D}(\lambda) := \mathbb{E}_{v \sim F_{(N)}}[\Psi_\lambda(v)].$$

**Lemma B.9** (Dual envelope). *For every $\lambda > 0$,*

$$\mathcal{D}(\lambda) = V_{(N)}(\lambda) + \lambda G_{(N)}(\lambda).$$

*Moreover,*

$$\inf_{\lambda > 0} \mathcal{D}(\lambda) = V_{(N)}(\lambda_\star).$$

*Proof.* Fix $v \in [0, B]$, $\lambda > 0$, and set

$$c_\lambda(v) := \left(1 + \tfrac{1}{\lambda}\right)v.$$

We can rewrite

$$\Phi_\lambda(v, b) = \mathbb{E}_{d^O \sim D}\left[\lambda(c_\lambda(v) - d^O)\mathbb{I}\{d^O < b\}\right].$$

Increasing $b$ up to $c_\lambda(v)$ only adds nonnegative contributions, whereas increasing $b$ beyond $c_\lambda(v)$ only adds nonpositive contributions. Hence $b = c_\lambda(v)$ maximizes $\Phi_\lambda(v, b)$, and

$$\Psi_\lambda(v) = \Phi_\lambda(v, c_\lambda(v)).$$

Taking expectation over $v \sim F_{(N)}$ yields

$$\mathcal{D}(\lambda) = V_{(N)}(\lambda) + \lambda G_{(N)}(\lambda).$$

It remains to prove the infimum identity. Consider the single-bidder primal problem

$$\text{OPT}_{\text{single}} := \sup_\pi \left\{\mathbb{E}[V^\pi] : \mathbb{E}[g^\pi] \geq 0\right\},$$

where the supremum is over measurable bidding rules $\pi : [0, B] \to \mathbb{R}_+$, and

$$V^\pi := v_{(N)}\mathbb{I}\{\pi(v_{(N)}) > d^O\}, \qquad g^\pi := (v_{(N)} - d^O)\mathbb{I}\{\pi(v_{(N)}) > d^O\}.$$

For any feasible $\pi$ and any $\lambda > 0$,
$$\mathbb{E}[V^\pi] \leq \mathbb{E}[V^\pi + \lambda g^\pi] \leq \mathcal{D}(\lambda).$$

Thus
$$\mathrm{OPT}_{\mathrm{single}} \leq \inf_{\lambda > 0} \mathcal{D}(\lambda).$$

If $\lambda_\star > 0$, then $G_{(N)}(\lambda_\star) = 0$. The bidding rule
$$\pi_\star(v) := \left(1 + \tfrac{1}{\lambda_\star}\right) v$$

is feasible and attains value $V_{(N)}(\lambda_\star)$. Also
$$\mathcal{D}(\lambda_\star) = V_{(N)}(\lambda_\star) + \lambda_\star G_{(N)}(\lambda_\star) = V_{(N)}(\lambda_\star).$$

Therefore
$$V_{(N)}(\lambda_\star) \leq \mathrm{OPT}_{\mathrm{single}} \leq \inf_{\lambda > 0} \mathcal{D}(\lambda) \leq \mathcal{D}(\lambda_\star) = V_{(N)}(\lambda_\star),$$

so equality holds throughout.

If $\lambda_\star = 0$, then $G_{(N)}(\lambda) \geq 0$ for every $\lambda > 0$, so $G_{(N)}(0^+) := \lim_{\lambda \downarrow 0} G_{(N)}(\lambda) \geq 0$. The always-win rule $\pi_0(v) \equiv 2B$ is feasible and attains value $\mathbb{E}[v_{(N)}] = V_{(N)}(0^+)$. Hence
$$V_{(N)}(0^+) \leq \mathrm{OPT}_{\mathrm{single}} \leq \inf_{\lambda > 0} \mathcal{D}(\lambda).$$

On the other hand, $|G_{(N)}(\lambda)| \leq B$, and therefore
$$\inf_{\lambda > 0} \mathcal{D}(\lambda) \leq \lim_{\lambda \downarrow 0} \left( V_{(N)}(\lambda) + \lambda G_{(N)}(\lambda) \right) = V_{(N)}(0^+).$$

Thus $\inf_{\lambda > 0} \mathcal{D}(\lambda) = V_{(N)}(0^+)$, which is $V_{(N)}(\lambda_\star)$ by definition. $\qquad \square$

**Lemma B.10** (Independent upper bound). *Under independent bidding,*
$$\limsup_{T \to \infty} \mathbb{E}\left[V_T^{I,A}\right] \leq V_{(N)}(\lambda_\star).$$

*Proof.* By Lemma B.5, $V_t^{I,A} \leq \bar{V}_t$ almost surely, so it suffices to upper bound the time-average value of the virtual bidder.

Fix $\lambda > 0$, and let $\mathcal{G}_t := \sigma(H_{t-1}, v_{1,t}, \ldots, v_{N,t})$ be the information after coalition values are realized but before the outside bid is drawn. Conditioning on $\mathcal{G}_t$, both $v_{(N),t}$ and $\bar{b}_t$ are fixed, while $d_t^O \sim D$ is fresh and independent. Hence
$$\mathbb{E}\left[\bar{V}_t + \lambda \bar{g}_t \mid \mathcal{G}_t\right] = \Phi_\lambda(v_{(N),t}, \bar{b}_t) \leq \Psi_\lambda(v_{(N),t}).$$

Taking expectations and using $v_{(N),t} \sim F_{(N)}$,
$$\mathbb{E}\left[\bar{V}_t + \lambda \bar{g}_t\right] \leq \mathcal{D}(\lambda).$$

Summing over $t = 1, \ldots, T$ and dividing by $T$,
$$\mathbb{E}\left[\frac{1}{T} \sum_{t=1}^T \bar{V}_t\right] \leq \mathcal{D}(\lambda) - \lambda \mathbb{E}\left[\frac{1}{T} \sum_{t=1}^T \bar{g}_t\right].$$

By Lemma B.8,
$$\sum_{t=1}^T \bar{g}_t \geq -N R_T \qquad \text{pathwise},$$

with $R_T = o(T)$. Therefore
$$\mathbb{E}\left[\frac{1}{T} \sum_{t=1}^T \bar{V}_t\right] \leq \mathcal{D}(\lambda) + \lambda \frac{N R_T}{T}.$$

Taking $\limsup_{T \to \infty}$ gives
$$\limsup_{T \to \infty} \mathbb{E}\left[\frac{1}{T} \sum_{t=1}^T \bar{V}_t\right] \leq \mathcal{D}(\lambda) \qquad \forall \lambda > 0.$$

Taking the infimum over $\lambda > 0$ and using Lemma B.9,
$$\limsup_{T \to \infty} \mathbb{E}\left[\frac{1}{T} \sum_{t=1}^T \bar{V}_t\right] \leq \inf_{\lambda > 0} \mathcal{D}(\lambda) = V_{(N)}(\lambda_\star).$$

Since $V_t^{I,A} \leq \bar{V}_t$ almost surely for every $t$, the same upper bound holds for $V_T^{I,A}$. $\qquad \square$

**Step 4: Conclusion**

Combining Lemma B.4 and Lemma B.10, we obtain

$$\liminf_{T\to\infty} \mathbb{E}\left[V_T^{C,A} - V_T^{I,A}\right] \geq V_{(N)}(\lambda_\star) - V_{(N)}(\lambda_\star) = 0.$$

This proves Theorem 4.1.

## B.2. Proof of Theorem 4.2

**Reduction to a single-bidder scenario.** To prove Theorem 4.2, we first draw a connection between the $N$-bidder coordination scenario and the scenario where a *single* bidder, whose value equals the highest value among the $N$ bidders, competes against the outside bid.

Formally, let $v_{(N)} = \max\{v_1, \dots, v_N\}$ denote the highest value among $N$ i.i.d. samples from $F$. Let $F_{(N)}$ be the distribution of $v_{(N)}$. Let $G_{(N)}(\lambda)$ be the expected utility of a single bidder with value $v_{(N)} \sim F_{(N)}$ and bidding using multiplier $\lambda > 0$, competing against the outside bid $d^O \sim D$:

$$G_{(N)}(\lambda) = \mathbb{E}_{v_{(N)}\sim F_{(N)}, d^O \sim D}\left[(v_{(N)} - d^O) \cdot \mathbb{I}\left[(1 + \tfrac{1}{\lambda})v_{(N)} > d^O\right]\right].$$

Let $G_i^C(\lambda_i)$ be the expected utility of bidder $i$ with value $v_i \sim F$ and multiplier $\lambda_i > 0$ in the $N$-bidder coordinated scenario:

$$G_i^C(\lambda_i) = \mathbb{E}_{v_1,\dots,v_N\sim F, d^O \sim D}\left[(v_i - d^O)x_i(\boldsymbol{v}, \lambda_i)\right]$$

where $x_i(\boldsymbol{v}, \lambda_i) = 1$ if $v_i = \max_{j\in[N]}\{v_j\}$ and $(1 + 1/\lambda_i)v_i > d^O$, and 0 otherwise.

We show that the $N$-bidder coordinated scenario is "equivalent" to the single-bidder scenario in the following sense:

**Observation B.1.** $G_i^C(\lambda_i) = \frac{1}{N}G_{(N)}(\lambda_i)$.

*Proof.* In the coordinated scenario, bidder $i$ obtains utility 0 if its value is not the highest among the $N$ bidders. The probability that bidder $i$ has the highest value is $1/N$ because the $N$ bidders' values are i.i.d. So,

$$\begin{aligned}
G_i^C(\lambda_i) &= 0 + \tfrac{1}{N}\mathbb{E}_{v_i\sim F|v_i=v_{(N)}}\left[(v_i - d^O)\mathbb{I}[(1 + 1/\lambda_i)v_i > d^O]\right] \\
&= \tfrac{1}{N}\mathbb{E}_{v_{(N)}\sim F_{(N)}}\left[(v_{(N)} - d^O)\mathbb{I}[(1 + 1/\lambda_i)v_{(N)} > d^O]\right] \\
&= \tfrac{1}{N}G_{(N)}(\lambda_i). \qquad\qquad\qquad\qquad\qquad\qquad\qquad\qquad\qquad\qquad\qquad\quad \square
\end{aligned}$$

Since the $N$-bidder coordinated scenario is equivalent to a single-bidder scenario, we can now analyze the single-bidder scenario.

In addition to Lemma B.1, we present the following useful lemma regarding the expected utility function $G_{(N)}(\lambda)$ of the single bidder, under Assumption 3.1.

**Lemma B.11.** *Under Assumption 3.1, $G_{(N)}(\lambda) \geq \Delta > 0$.*

*Proof.* By Lemma B.1, $G_{(N)}(\lambda)$ is increasing in $\lambda$. So,

$$\begin{aligned}
G_{(N)}(\lambda) &\geq \lim_{\mu\to 0^+} G_{(N)}(\mu) \\
&= \mathbb{E}_{v_{(N)}\sim F_{(N)}, d^O\sim D}\left[v_{(N)} - d^O\right] \\
&= \mathbb{E}\left[(v_{(N)} - d^O)_+\right] - \mathbb{E}\left[(d^O - v_{(N)})_+\right] \\
&\geq \mathbb{E}\left[(v_{(N-1)} - d^O)_+\right] - \mathbb{E}\left[(d^O - v_{(N)})_+\right] \\
&\geq \Delta > 0
\end{aligned}$$

where the last "$\geq$" follows from Assumption 3.1. $\qquad\qquad\qquad\qquad\qquad\qquad\qquad\qquad\quad \square$

Intuitively, Lemma B.11 shows that the expected utility obtained by the highest-value coalition bidder is always positive (by a margin of $\Delta > 0$). This means that the bidder's multiplier $\lambda_t$, following mirror descent updates, will decrease in expectation every round and eventually converge to 0. We formalize this intuition below.

**Coordinated multipliers converge to** $0$**.** Using the above characterizations, we show an interesting fact: the multipliers $\lambda_{i,t}^C$ of auto-bidders running Algorithm 3 in the coordinated scenario *converge to* $0$ *as* $t \to \infty$. This means that the coalition will eventually submit arbitrarily high bids $b_{i,t}^C = (1 + \frac{1}{\lambda_{i,t}^C})v_{i,t} \to \infty$.

**Lemma B.12.** *Under Assumption 3.1, for any* $i \in [N], t \in [T]$*, with probability at least* $1 - \exp\left(-\frac{\Delta^2 t}{32B^2N^2}\right)$*,* $h'(\lambda_{i,t+1}^C) \leq h'(\lambda_{i,1}) - \frac{\Delta t}{2N\sqrt{T}}$*.*

*Proof.* Let $y_{i,t} := h'(\lambda_{i,t}^C)$. According to Algorithm 3 and Equation (4),

$$y_{i,t+1} = y_{i,t} - \alpha g_{i,t} \tag{9}$$

where each $g_{i,t}$ is an unbiased estimator of bidder $i$'s expected utility $G_i^C(\lambda_{i,t}^C) = \mathbb{E}[g_{i,t}]$ at round $t$. If bidder $i$ is not the highest-value bidder in round $t$, then $g_{i,t} = 0$, meaning that the bidder's multiplier $\lambda_{i,t}^C$ is not updated in that round. According to Observation B.1 and Lemma B.11, bidder $i$'s expected utility is at least

$$\mathbb{E}[g_{i,t'}] = G_i^C(\lambda_{i,t'}^C) = \tfrac{1}{N}G_{(N)}(\lambda_{i,t'}) \geq \tfrac{\Delta}{N}.$$

Because values and outside bids are bounded by $B$, the realized utility $g_{i,t'} \in [-B, B]$. So, by Azuma's inequality:

$$\Pr\left[\sum_{t'=1}^t g_{i,t'} < t \cdot \tfrac{\Delta}{N} - \varepsilon\right] \leq \exp\left(-\tfrac{\varepsilon^2}{8B^2t}\right).$$

Let $\varepsilon = \frac{t\Delta}{2N}$. We have with probability at least $1 - \exp(-\frac{t\Delta^2}{32B^2N^2})$,

$$y_{i,t+1} \leq y_{i,1} - \alpha\left(t \cdot \tfrac{\Delta}{N} - \varepsilon\right) = y_{i,1} - \tfrac{t\Delta}{2N\sqrt{T}}.$$

where we plugged in $\alpha = 1/\sqrt{T}$. $\qquad\square$

**Coordinated total value is larger than independent total value.** Using the fact that coordinated multipliers converge to $0$, we show that the total value obtained by coordinated auto-bidders is larger than that under independent bidding in the limit $T \to \infty$.

Let $V_{(N)}(\lambda)$ be the expected value obtained by a single bidder whose value is the highest value $v_{(N)}$ among $N$ samples from $F$ and who uses multiplier $\lambda > 0$ to bid against outside bid $d^O \sim D$:

$$V_{(N)}(\lambda) := \mathbb{E}_{v_{(N)} \sim F_{(N)}, d^O \sim D}\left[v_{(N)}\mathbb{I}\left[(1 + \tfrac{1}{\lambda})v_{(N)} > d^O\right]\right]. \tag{10}$$

Because $V_{(N)}(\lambda)$ is monotone in $\lambda$ and bounded, the limit

$$V_{(N)}(0^+) := \lim_{\lambda \to 0^+} V_{(N)}(\lambda)$$

exists. Note that the expected total value of the $N$ coordinated bidders with multipliers $\lambda_1^C, \ldots, \lambda_N^C$ is

$$V^C(\lambda_1^C, \ldots, \lambda_N^C) = \mathbb{E}\left[v_{i^*}\mathbb{I}\left[(1 + \tfrac{1}{\lambda_{i^*}})v_{i^*} > d^O\right]\right] = \mathbb{E}\left[V_{(N)}(\lambda_{i^*}^C)\right]$$

where $i^* = \arg\max_{i \in [N]} v_i$. This means that the expected average total value of the coordinated bidders during $T$ rounds is

$$\mathbb{E}\left[\frac{1}{T}\sum_{t=1}^T V_t^{C,A}\right] = \mathbb{E}\left[\frac{1}{T}\sum_{t=1}^T V_{(N)}(\lambda_{i_t^*,t}^C)\right] \tag{11}$$

where $i_t^* \in [N]$ is the highest-value bidder in round $t$.

**Lemma B.13.** *For any coordination mechanism* $\mathcal{G}$ *and any round* $t$*,*

$$\mathbb{E}[V_t^{\mathcal{G}}] \leq V_{(N)}(0^+) = \mathbb{E}_{v_{(N)} \sim F_{(N)}}[v_{(N)}].$$

*Consequently, for every* $T \geq 1$*,*

$$\mathbb{E}\left[\frac{1}{T}\sum_{t=1}^T V_t^{\mathcal{G}}\right] \leq V_{(N)}(0^+).$$

*Proof.* Fix a round $t$ and write $v_{(N),t} := \max_{i \in [N]} v_{i,t}$. In a single-item auction, the coalition's realized value is either 0 (if the outside bid wins) or equals the realized value of the winning coalition bidder, which is always at most $v_{(N),t}$. Hence $V_t^{\mathcal{G}} \leq v_{(N),t}$ almost surely, and taking expectations gives $\mathbb{E}[V_t^{\mathcal{G}}] \leq \mathbb{E}[v_{(N),t}]$. Finally, since bids and $d^O$ are bounded, $(1 + \frac{1}{\lambda})v_{(N)} \to +\infty$ as $\lambda \downarrow 0$, so the single bidder wins with probability 1 in the limit and thus $V_{(N)}(0^+) = \mathbb{E}[v_{(N)}]$. $\square$

**Proof of Theorem 4.2.** By Lemma B.12, with constant $c := \frac{\Delta^2}{32B^2N^2}$, for any fixed $i$ and $t \geq 1$, the coordinated multiplier $\lambda_{i,t+1}^C$ satisfies

$$\Pr\left[h'(\lambda_{i,t+1}^C) \leq h'(\lambda_{i,1}) - \frac{t\Delta}{2N\sqrt{T}}\right] \geq 1 - e^{-ct}. \tag{12}$$

Fix $\varepsilon > 0$. Since $V_{(N)}(0^+)$ exists, there is $\delta = \delta(\varepsilon) > 0$ such that

$$\left|V_{(N)}(0^+) - V_{(N)}(\lambda)\right| < \varepsilon \quad \text{for all } \lambda \in (0, \delta]. \tag{13}$$

Since $h'$ is strictly increasing and $\lim_{\lambda \downarrow 0} h'(\lambda) = -\infty$, we can define $b(\delta) := (h')^{-1}(\delta)$.

Define

$$m_T := \left\lceil \frac{2N\sqrt{T}}{\Delta}(h'(\lambda_{i,1}) - b(\delta)) \right\rceil = O\left(\sqrt{T}(-b(\delta))\right),$$

so for all $t \geq m_T$,

$$h'(\lambda_{i,1}) - \frac{t\Delta}{2N\sqrt{T}} \leq h'(\lambda_{i,1}) - \frac{m_T\Delta}{2N\sqrt{T}} \leq b(\delta).$$

Let $E_{i,t}$ be the event inside the probability in (12). A union bound over $i = 1, \ldots, N$ and $t \geq m_T$ gives

$$\Pr\left[\bigcap_{i=1}^{N} \bigcap_{t=m_T}^{T} E_{i,t}\right] \geq 1 - NTe^{-cm_T} =: 1 - p_T. \tag{14}$$

On this event, for all $t \geq m_T$ we have $h'(\lambda_{i_t^*,t}^C) \leq b(\delta)$, which is equivalent to $\lambda_{i_t^*,t}^C \leq \delta$, hence by Eq. (13),

$$\left|V_{(N)}(0^+) - V_{(N)}(\lambda_{i_t^*,t}^C)\right| < \varepsilon.$$

For the first $m_T$ rounds we have $|V_{(N)}(0^+) - V_{(N)}(\cdot)| \leq B$ due to bounded value. Therefore, conditioning on the event in (14),

$$\left|V_{(N)}(0^+) - \frac{1}{T}\sum_{t=1}^{T} V_{(N)}(\lambda_{i_t^*,t}^C)\right| \leq \frac{m_T}{T}B + \frac{T - m_T}{T}\varepsilon.$$

Taking expectations and using that the per-round gap is $\leq B$ on the complement of the event in (14), we have

$$\mathbb{E}\left[\left|V_{(N)}(0^+) - \frac{1}{T}\sum_{t=1}^{T} V_{(N)}(\lambda_{i_t^*,t}^C)\right|\right]$$
$$\leq \frac{m_T}{T}B + \frac{T - m_T}{T}\varepsilon + B\,p_T$$
$$\leq \frac{O(\sqrt{T}(-b(\delta)))}{T}B + \varepsilon + BNTe^{-cO(\sqrt{T}(-b(\delta)))} \leq 3\varepsilon$$

for sufficiently large $T$. So we obtain

$$\lim_{T \to \infty} \mathbb{E}\left[\frac{1}{T}\sum_{t=1}^{T} V_{(N)}(\lambda_{i_t^*,t}^C)\right] = V_{(N)}(0^+).$$

Combining with Lemma B.13, we conclude that

$$V_{(N)}(0^+) \geq \mathbb{E}\left[\frac{1}{T}\sum_{t=1}^{T} V_t^{\mathcal{G}}\right],$$

which proves Theorem 4.2. $\square$

## C. Omitted Proofs in Section 5

### C.1. Proof of Theorem 5.1

The argument parallels the proof of Theorem 3.1.

Let $i_t^* = \arg\max_{i \in [N]} v_{i,t}$ denote the index of the highest-value bidder in round $t$. Conditioned on the history $H_{t-1}$, we analyze the expected difference $\Delta_{i,t} = u_{i,t}^{C,A} - u_{i,t}^{\text{Truth}}$ for bidder $i$:

$$
\begin{aligned}
&\mathbb{E}_{\boldsymbol{v}_t \sim F, \, d_t^O \sim D}\big[\Delta_{i,t} \mid H_{t-1}\big] \\
&= \mathbb{E}_{\boldsymbol{v}_t, \, d_t^O}\big[\mathbb{I}[i_t^* = i] \cdot \Delta_{i,t} \mid H_{t-1}\big] \\
&= \mathbb{E}\Big[\mathbb{I}[i_t^* = i] \cdot \mathbb{I}[v_{i,t} \geq d_t^O] \cdot (d_t^{\text{Truth}} - d_t^O)\Big] + \mathbb{E}\Big[\mathbb{I}[i_t^* = i] \cdot \mathbb{I}[v_{i,t} \leq d_t^O \leq b_{i,t}^{C,A}] \cdot (v_{i,t} - d_t^O)\Big] \\
&= \mathbb{E}\Big[\mathbb{I}[i_t^* = i] \cdot (d_t^{\text{Truth}} - d_t^O)\Big] + \mathbb{E}\Big[\mathbb{I}[i_t^* = i] \cdot \mathbb{I}[v_{i,t} \leq d_t^O \leq b_{i,t}^{C,A}] \cdot (v_{i,t} - d_t^O)\Big] \\
&\geq \mathbb{E}\Big[\mathbb{I}[i_t^* = i] \big(\max_{j \neq i} v_{j,t} - d_t^O\big)_+\Big] + \mathbb{E}\Big[\mathbb{I}[i_t^* = i] \cdot \mathbb{I}[v_{i,t} \leq d_t^O] \cdot (v_{i,t} - d_t^O)\Big].
\end{aligned}
$$

Summing over all bidders $i \in [N]$, the indicator $\mathbb{I}[i_t^* = i]$ ensures that only the highest-value bidder contributes in each round. Hence,

$$
\mathbb{E}\Big[\sum_{i=1}^N \Delta_{i,t} \mid H_{t-1}\Big] \geq \mathbb{E}\big[(v_{(N-1)} - d_t^O)_+\big] - \mathbb{E}\big[(d_t^O - v_{(N)})_+\big]
$$

$$
\geq \Delta,
$$

where the second inequality follows from Assumption 3.1. Summing over $t \in [T]$ gives $\mathbb{E}[\sum_{i=1}^N U_{i,T}^{C,A} - \sum_{i=1}^N U_{i,T}^{I,A}] \geq \Delta T$.

### C.2. Proof of Theorem 5.2

The argument parallels the proof of Theorem 4.2.

Let $G_i^C(\lambda_i)$ denote bidder $i$'s expected utility when $v_i \sim F_i$ and bidder $i$ uses multiplier $\lambda_i > 0$ in the coordinated scenario:

$$
G_i^C(\lambda_i) \;=\; \mathbb{E}_{v_1 \sim F_1, \ldots, v_N \sim F_N, \, d^O \sim D}\big[(v_i - d^O)\, x_i(\boldsymbol{v}, \lambda_i)\big],
$$

where $x_i(\boldsymbol{v}, \lambda_i) = 1$ if $v_i = \max_{j \in [N]}\{v_j\}$ and $(1 + \frac{1}{\lambda_i})v_i > d^O$, and 0 otherwise.

By reusing the proof of Lemma B.1 with $D$ replaced by the distribution of $\max\{d^O, \max_{j \neq i} v_j\}$, we claim that $G_i^C(\lambda_i)$ is *increasing* in $\lambda_i$. Hence,

$$
G_i^C(\lambda_i) \;\geq\; \lim_{\mu \to 0^+} G_i^C(\mu) \;=\; E\Big[(v_i - d^O)\, \mathbb{I}\{v_i = \max_{j \in [N]}\{v_j\}\}\Big] \;=:\; \Delta_i,
$$

where the limit exists because $G_i^C(\cdot)$ is bounded and monotone.

We now present the parallel version of Lemma B.12.

**Lemma C.1.** *Under Assumption 5.1, for any $i \in [N]$ and $t \geq 1$, with probability at least $1 - \exp\big(-\frac{\Delta_i^2 t}{32B^2}\big)$,*

$$
h'(\lambda_{i,t+1}^C) \leq h'(\lambda_{i,1}) - \frac{\Delta_i t}{2\sqrt{T}}.
$$

*Proof.* The expected utility $\mathbb{E}[g_{i,t'}]$ is lower bounded by

$$
\mathbb{E}[g_{i,t'}] \;=\; G_i^C(\lambda_{i,t'}) \;\geq\; \Delta_i.
$$

Applying Azuma's inequality (exactly as in the proof of Lemma B.12) completes the argument. $\qquad\square$

Write $v_{(N)} := \max_{j \in [N]} v_j$ with $v_j \sim F_j$ mutually independent. Its CDF is $F^{\max}(x) = \Pr[v_{(N)} \le x] = \prod_{j=1}^{N} F_j(x)$. For $\lambda > 0$, define

$$V_{(N)}(\lambda) := \mathbb{E}_{v_{(N)} \sim F^{\max}, \, d^O \sim D}\Big[ v_{(N)} \cdot \mathbb{I}\{(1 + \tfrac{1}{\lambda}) v_{(N)} > d^O\} \Big],$$
$$V_{(N)}(0^+) := \lim_{\lambda \to 0^+} V_{(N)}(\lambda) = \mathbb{E}_{v_{(N)} \sim F^{\max}}\big[ v_{(N)} \big].$$

The limit $V_{(N)}(0^+) = \lim_{\lambda \to 0^+} V_{(N)}(\lambda)$ exists because $V_{(N)}(\lambda)$ is monotone and bounded. In the i.i.d. special case $F_1 = \cdots = F_N = F$, $F^{\max}$ coincides with the order-statistic distribution $F_{(N)}$, recovering the original definition.

In the non-i.i.d. case, Lemma B.13 still holds:

**Lemma C.2** (copy of Lemma B.13). *For any coordination mechanism $\mathcal{G}$ and any round $t$,*

$$\mathbb{E}\big[V_t^{\mathcal{G}}\big] \le V_{(N)}(0^+) = \mathbb{E}_{v_{(N)} \sim F^{\max}}\big[v_{(N)}\big].$$

*Consequently, for every $T \ge 1$,*

$$\mathbb{E}\Big[\frac{1}{T} \sum_{t=1}^{T} V_t^{\mathcal{G}}\Big] \le V_{(N)}(0^+).$$

**Proof of Theorem 5.2.** Fix $\varepsilon > 0$ and choose $\delta = \delta(\varepsilon) > 0$ as in Equation (13). Let

$$\Delta_{\min} := \min_{i \in [N]} \Delta_i > 0 \qquad c_{\min} := \frac{\Delta_{\min}^2}{32 B^2} > 0.$$

Define $b(\delta) := (h')^{-1}(\delta)$ and

$$m_T := \Big\lceil \frac{2\sqrt{T}}{\Delta_{\min}}(h'(\lambda_{i,1}) - b(\delta)) \Big\rceil.$$

By Lemma C.1, for each $i$ and every $t \ge m_T$, $\lambda_{i,t+1}^C \le \delta$ holds with probability at least $1 - \exp(-c_{\min} t)$. A union bound over $i = 1, \ldots, N$ and $t = m_T, \ldots, T$ yields

$$\Pr\Big[\bigcap_{i=1}^{N} \bigcap_{t=m_T}^{T} \{\lambda_{i,t+1}^C \le \delta\}\Big] \ge 1 - NT e^{-c_{\min} m_T} =: 1 - p_T.$$

On this event, for all $t \ge m_T$ we have $\lambda_{i_t^*, t}^C \le \delta$, hence by (13),

$$\big|V_{(N)}(0^+) - V_{(N)}(\lambda_{i_t^*, t}^C)\big| < \varepsilon,$$

where $i_t^* = \arg\max_{i \in [N]} v_{i,t}$. For each of the first $m_T$ rounds, the absolute gap is $\le B$. Therefore,

$$\Big|V_{(N)}(0^+) - \frac{1}{T} \sum_{t=1}^{T} V_{(N)}(\lambda_{i_t^*, t}^C)\Big| \le \frac{m_T}{T} B + \frac{T - m_T}{T} \varepsilon$$

on the above high-probability event; on its complement the gap is $\le B$. Taking expectations and using $p_T \le NT e^{-c_{\min} m_T}$ gives

$$\mathbb{E}\Big[\Big|V_{(N)}(0^+) - \frac{1}{T} \sum_{t=1}^{T} V_{(N)}(\lambda_{i_t^*, t}^C)\Big|\Big] \le \frac{m_T}{T} B + \varepsilon + B \, p_T \xrightarrow[T \to \infty]{} 0.$$

Hence

$$\lim_{T \to \infty} \mathbb{E}\Big[\frac{1}{T} \sum_{t=1}^{T} V_{(N)}(\lambda_{i_t^*, t}^C)\Big] = V_{(N)}(0^+).$$

Finally, combining with Lemma C.2 yields the desired inequality against the independent-bidding total value, completing the proof. $\qquad\square$

# D. Additional Experiment Results

This appendix presents additional figures in the experiments: Figure 4 for the symmetric setting on synthetic data; Figure 5 for the asymmetric setting on synthetic data; Figure 6 for the real-world dataset.

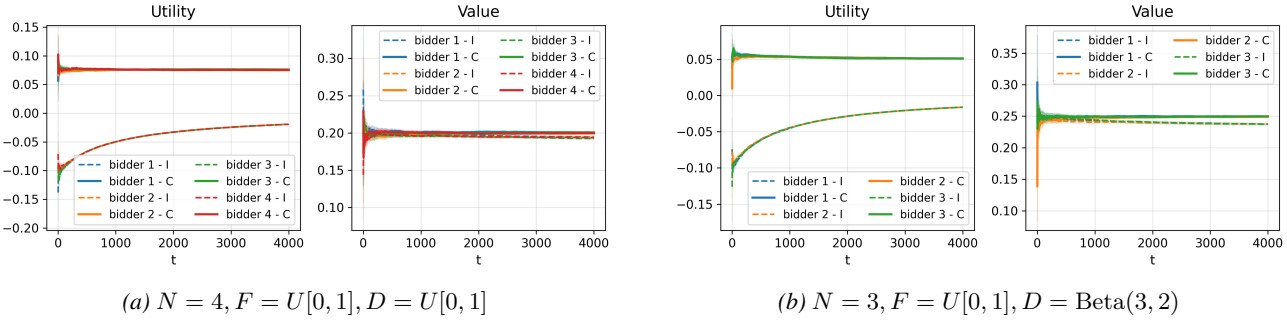

*(a)* $N = 4, F = U[0,1], D = U[0,1]$        *(b)* $N = 3, F = U[0,1], D = \mathrm{Beta}(3,2)$

*Figure 4.* Experiments under i.i.d auto-bidders.

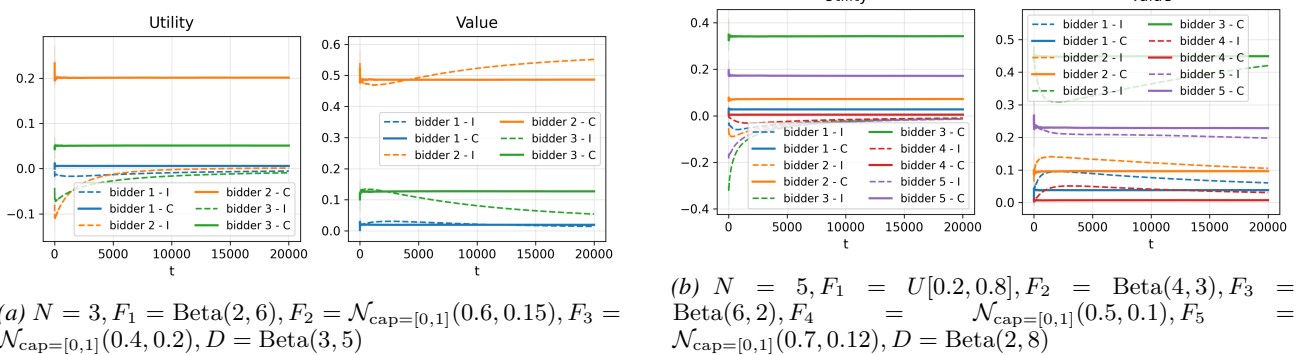

*(a)* $N = 3, F_1 = \mathrm{Beta}(2,6), F_2 = \mathcal{N}_{\mathrm{cap}=[0,1]}(0.6, 0.15), F_3 = \mathcal{N}_{\mathrm{cap}=[0,1]}(0.4, 0.2), D = \mathrm{Beta}(3,5)$

*(b)* $N = 5, F_1 = U[0.2, 0.8], F_2 = \mathrm{Beta}(4,3), F_3 = \mathrm{Beta}(6,2), F_4 = \mathcal{N}_{\mathrm{cap}=[0,1]}(0.5, 0.1), F_5 = \mathcal{N}_{\mathrm{cap}=[0,1]}(0.7, 0.12), D = \mathrm{Beta}(2,8)$

*Figure 5.* Experiments under non-i.i.d auto-bidders.

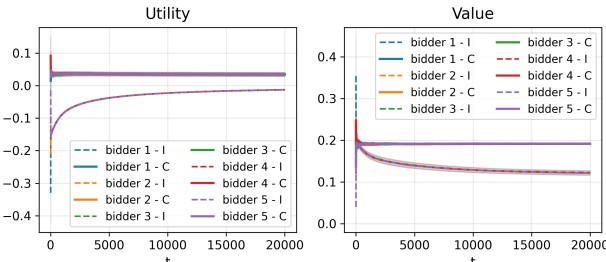

*Figure 6.* Experiments in real-world datasets. ($N = 5$).

