# OpenReview forum: "On the Coordination of Value-Maximizing Bidders"
_ICML.cc/2026/Conference — ICML 2026 regular_

### Official Review · Reviewer_zgLj · 2026-03-09

**Soundness:** 3
**Presentation:** 3
**Significance:** 2
**Originality:** 3
**Overall Recommendation:** 4
**Confidence:** 3

**Summary:**

This paper investigates the problem of strategic collaborative bidding among multiple auto-bidders in online advertising platforms. It is the first work, under the value-maximizer setting, to establish that a collaborative mechanism — in which the highest bidder among the coalition competes against external bidders while the remaining coalition members abstain from the auction — can improve ROS compliance and increase the total value accrued by participating auto-bidders. Synthetic and real-world datasets are employed to experimentally validate the theoretical findings.

**Compliance With Llm Reviewing Policy:**

Affirmed.

**Final Justification:**

My initial concerns have been fully addressed by the rebuttals, which strengthens my confidence in the paper's practical value. I maintain my weak accept recommendation.

**Key Questions For Authors:**

Q1. As described in Algorithm 2, the proposed collaborative mechanism presupposes the existence of a planner with access to the private valuations of all coalition members. Is this assumption realistic in practice, and could coalition members have strategic incentives to misreport their true valuations? This concern may confine the theoretical setting to the narrower scenario of strategic bidding among multiple campaigns managed by a single advertiser.
Q2. The paper assumes that the external bidding environment is i.i.d. If external bidders also adopt similar strategic bidding algorithms, would the additional benefits of collaboration be diminished? A preliminary discussion or experimental simulation addressing this question would strengthen the practical relevance of the paper.
Q3. The paper focuses exclusively on mirror-descent-based auto-bidding algorithms. Can the proposed theoretical or empirical results be generalized to other auto-bidding algorithm families?

**Limitations:**

The authors are encouraged to add a dedicated subsection within the Discussion section to explicitly address the limitations of the theoretical framework and experimental results, including simplifying assumptions introduced in the problem setup and constraints in the empirical experimental design.

**Strengths And Weaknesses:**

Strengths
S1. The paper is technically sound, with rigorous theoretical analysis, and the experimental results provide solid support for the theoretical findings in the i.i.d. setting.
S2. The paper is well-written and well-structured, with clear and precise exposition.
S3. Existing auto-bidding research predominantly focuses on optimizing advertising value and utility from the perspective of independent bidders. This paper explores the relatively understudied setting of collaborative bidding strategies and, for the first time, derives meaningful theoretical results under the value-maximizer setting, demonstrating notable significance and originality.

Weaknesses
W1. The theoretical results rely on relatively simplified assumptions — for instance, external bids are assumed to be i.i.d., without accounting for the possibility that external bidders may adopt analogous collaborative strategies. Furthermore, the analysis is restricted to single-item auctions, which limits the direct applicability of the theoretical results to real-world online advertising environments.
W2. The evaluation on real-world data is considerably limited, as it relies on a single empirical i.i.d. model, providing insufficient support for results beyond the i.i.d. regime.
W3. The fairness implications in the extended non-i.i.d. setting receive inadequate discussion. Specifically, if a coalition member has a persistently low probability of being the highest bidder, the proposed mechanism would consistently require that member to abstain from bidding, raising potential fairness concerns.

---

> ### Author Rebuttal · Authors · 2026-03-30
>
> > (W2)  The evaluation on real-world data is limited, as it relies on a single empirical i.i.d. model.
>
> We used an empirical i.i.d. model due to the limitation of the iPinYou dataset: it lacks persistent bidder identities, which makes constructing a reliable non-i.i.d. benchmark challenging.  Prior work using the same dataset (e.g., Chen et al., 2023) also adopts an empirical i.i.d. model.
>
>
> > (W3) The fairness implications in the extended non-i.i.d. setting receive inadequate discussion.  Specifically, if a coalition member has a persistently low probability of being the highest bidder, the proposed mechanism would consistently require that member to abstain from bidding.
>
> “Unfairness” is inherent to auctions: a bidder with persistently lower value should win with low probability even under independent bidding. Coordination is not meant to resolve the unfairness in auctions, but rather avoids inefficient internal competition and improves total value.  And as we discussed after Theorem 5.1, the resulting gain in total value can be redistributed by the coordinator to the bidders within the coalition (by monetary transfers), which may help address fairness concerns in practice.
>
> > (Q1) As described in Algorithm 2, the proposed collaborative mechanism presupposes the existence of a planner with access to the private valuations of all coalition members. Is this assumption realistic in practice, and could coalition members have strategic incentives to misreport their true valuations? This concern may confine the theoretical setting to the narrower scenario of strategic bidding among multiple campaigns managed by a single advertiser.
>
> In ad auction practice, the assumption that the planner has access to the values of advertisers is reasonable.  The planner is the autobidding platform that can estimate an ad’s probability of click or conversion (see, e.g., [Google’s auto-bidding service](https://support.google.com/google-ads/answer/2979071?hl=en)). The advertisers themselves, on the other hand, might be the ones who do not know the values of their ads.  So we think our theoretical setting, not only capturing multiple campaigns by a single advertiser, applies to some scenarios with multiple advertisers as well.
>
>
> > (Q2 & W1) The paper assumes that the external bidding environment is i.i.d. If external bidders also adopt similar strategic bidding algorithms, would the additional benefits of collaboration be diminished?
>
>
> We include additional experiments where the external bidders are also strategic (also using mirror-descent auto-bidding algorithms). We have $N$ coalition bidders and $O$ outside bidders. All outside bidders use the same mirror-descent auto-bidding algorithm as independent bidders.
>
> We observe that coordination continues to improve total value over independent bidding:
>
> | # | Regime | $N$ | $O$ | Rounds | Total Utility (I) | Total Utility (C) | Total Value (I) | Total Value (C) |
> |---|--------|---------------|-------------|--------|--------------|--------------|----------|----------|
> | 1 | i.i.d. | 2 | 1 | 4000 | -0.022 ± 0.000 | 0.014 ± 0.001 | 0.500 ± 0.002 | 0.603 ± 0.002 |
> | 2 | i.i.d. | 4 | 2 | 4000 | -0.089 ± 0.000 | -0.031 ± 0.000 | 0.572 ± 0.002 | 0.680 ± 0.002 |
> | 3 | i.i.d. | 3 | 3 | 4000 | -0.067 ± 0.000 | -0.048 ± 0.000 | 0.428 ± 0.001 | 0.491 ± 0.002 |
> | 4 | non-i.i.d. | 2 | 1 | 10000 | -0.002 ± 0.000 | 0.015 ± 0.001 | 0.681 ± 0.001 | 0.696 ± 0.001 |
> | 5 | non-i.i.d. | 3 | 2 | 20000 | -0.025 ± 0.000 | -0.009 ± 0.000 | 0.488 ± 0.001 | 0.550 ± 0.001 |
> | 6 | non-i.i.d. | 5 | 3 | 20000 | -0.064 ± 0.000 | -0.007 ± 0.000 | 0.739 ± 0.001 | 0.792 ± 0.001 |
>
>
> The value distributions of the coalition bidders are identical to the six synthetic settings in the paper. The outside bidders’ value distributions in each setting are:
> - i.i.d. settings: $U[0,1]$.
> - non-i.i.d. settings:
>   - Setting 4: $U[0.2, 0.8]$.
>   - Setting 5: $\mathrm{Beta}(3,5)$ and $U[0.25, 0.75]$.
>   - Setting 6: $\mathrm{Beta}(2,8)$, $U[0.1, 0.6]$, and truncated normal $\mathcal{N}(0.55, 0.12^2)$ clipped to $[0,1]$.
>
> These results provide empirical evidence that the benefit of coordination is robust to strategic outside bidders.
>
>
> > (Q3) The paper focuses exclusively on mirror-descent-based auto-bidding algorithms. Can the proposed theoretical or empirical results be generalized to other auto-bidding algorithm families?
>
> We clarify that our results are not limited to mirror-descent-based algorithms. In particular, the RoS guarantees (Theorem 3.1) apply to any overbidding auto-bidding algorithm.
>
> For the value guarantees (Theorems 4.1 and 4.2), we focus on mirror-descent-based algorithms because they are the state-of-the-art approach for auto-bidding under RoS constraints and achieve the best known theoretical guarantees.  Moreover, Theorem 4.2 discusses other coordination mechanisms and shows that coordinated mirror descent is asymptotically optimal among all coordination mechanisms.

---

> > ### Author Rebuttal · Reviewer_zgLj · 2026-04-03
> >
> > Thank you for the constructive rebuttal. My initial concerns have been fully addressed, which strengthens my confidence in the paper's practical value. After reviewing the clarifications, I maintain my original score of Weak Accept, and am pleased to see this paper accepted.

---

### Official Review · Reviewer_aUCD · 2026-03-12

**Soundness:** 3
**Presentation:** 3
**Significance:** 2
**Originality:** 3
**Overall Recommendation:** 4
**Confidence:** 3

**Summary:**

This work considers an an online auto-bidding scenario, where a coalition of bidders coordinate between themselves in a second-price auction to increase their total value and return-on-spend. Every bidder aims at maximizing the accumulated value while keeping a positive utility (value minus payed costs), i.e., a positive return on spend (RoS). At every round, only the bidder with the highest valuation in the coalition participate in the auction, while the others bid zero. The actual bid is computed by an overbidding algorithm.

The authors first address the scenario where every bidder of the coalition draw their valuation from the same distribution. Under an appropriate assumption, they show that every bidder increases their own expected utility, thus reducing the RoS constraint violation.

They then study the case where every bidder runs a mirror-descend algorithm, and show that asymptotically their coordination mechanism weakly increases the total value obtained by the coalition and is weakly better than other coordinated bidding strategy.

When the valuations of the coalition are not i.i.d., the paper shows that the overall RoS of the coalition is increased in expectation. Furthermore, if every bidder is competitive enough, the value accumulated by the coalition is asymptotically better.

**Compliance With Llm Reviewing Policy:**

Affirmed.

**Final Justification:**

The paper is technically solid, but its results are not very significant. Many results hold only asymptotically. Furthermore, it remains unclear when the additional coordination mechanism strictly increases the total value.

**Key Questions For Authors:**

- Algorithm 3 is actually a family of Algorithms, depending on the mirror-descend based algorithm A, and is a subset of all the possible coordination mechanisms.  Does the theorem claim that for any A its total accumulated value in expectation is asymptotically at least the value of of any other mechanism, possibly even other mirror-descend based mechanisms?
- When a bidder bids independently and its multipliers converge to zero, I would expect a large violation of the RoS constraint, as the bidder tends to win even when the competing bid is larger than their valuation. Why does your result at lines 316-317 not lead the coalition to increase their RoS violation?  Is it because thanks to Assumption 3.1, the single bidder of the coalition that actually participate in the auctions has valuation larger than the competing bid in expectation?
- Why do the results on the bidder utility (e.g. Theorem 3.1) show that it increases of at least a strictly positive quantity in finite time, while those on the accumulated value show that asymptotically it is no worse than that attained without a coordination mechanism? Is it possible to show that coordination strictly increases the value for sufficiently large T?

**Limitations:**

yes

**Strengths And Weaknesses:**

**Strengths.** The paper addresses a novel and well-motivated problem. It shows how coalitions of bidders can cooperate to increase their performances by analyzing the different metrics typically employed in the literature. To the best of my knowledge, this is the first work to consider such a problem. While existing works focus on cases where multiple bidders reach some sort of cooperation or collusion at an equilibrium, here the bidders belong to the same entity and such an equilibrium concept is not necessary.
The paper provides multiple general results with few assumptions, with a clear presentation of the theoretical results for both i.i.d and non i.i.d bidders. The proof sketches and intuitions provided in the main paper seem sound. Empirical experiments validate the theoretical results.

**Weaknesses.** The main weakness concern the results on the coalition total value, specifically Theorem 4.1 and Theorem 5.2. These two results consider the total value of the coalition, rather than individual bidders. Hence, a bidder may loose value with respect to an individual bidding scenario. Furthermore, they do not quantify the increment in the accumulated value, which thus may be small or null, and they hold only asymptotically.

The second part of Theorem 3.1, namely that Assumption 3.1 is necessary, is also a bit weak. It basically shows that when the assumption is violated, there is an overbidding algorithm A such that every bidder would more utility using A without coordination than using A with the coordination mechanism. However, this is a very precise Algorithm A that bids always the bidder's valuation when used without the coalition, and instead may bid 1 when coordinating with the others. This is not particularly interesting, as we can choose the overbidding algorithm and we would employ a mirror-descend algorithm or at least one that achieves sublinear regret and violation.
Nonetheless, the fact that the assumption is satisfied for sufficiently large coalitions already justify it.

Finally, I find Theorem 4.2 slightly unclear (see also questions).

---

> ### Author Rebuttal · Authors · 2026-03-30
>
> > (Weakness 1) The main weakness concern the results on the coalition total value, specifically Theorem 4.1 and Theorem 5.2.
>
> Theorems 4.1 and 5.2 indeed focus on coalition-level value and do not guarantee improvement for every individual bidder. This is probably unavoidable in the non-iid case (Theorem 5.2), because in the extreme case where one bidder is dominated by another (consistently having lower value), a reasonable coordination mechanism should never select the dominated bidder, so the dominated bidder may obtain lower value under coordination than independent bidding.  However, this limitation can be mitigated in practice by allowing the coordinator to redistribute the total gain among the bidders in the coalition (by monetary transfers) to improve the profit of everyone, as we already discussed after Theorem 5.1.
>
>
> > (Question 1 & Weakness 3) I find Theorem 4.2 slightly unclear.
> Algorithm 3 is actually a family of Algorithms, depending on the mirror-descend based algorithm A, and is a subset of all the possible coordination mechanisms. Does Theorem 4.2 claim that for any A its total accumulated value in expectation is asymptotically at least the value of of any other mechanism, possibly even other mirror-descend based mechanisms?
>
> Mirror-descent-based algorithms are indeed a subset of all possible coordination mechanisms.  Theorem 4.2 claims that **any** mirror-descent-based algorithm $A$ is _asymptotically optimal_ among all coordination mechanisms – namely, its total expected value is asymptotically at least that of any other mechanism, and any two mirror-descent-based algorithms obtain the same value asymptotically.  We will rephrase Theorem 4.2 to “any coordinated mirror-descent algorithm $A$ (Algorithm 3) is asymptotically optimal among all coordination mechanisms: namely, for any $\mathcal G$, the expected value of coordinated A is weakly larger than the expected value of $\mathcal G$ as $T\to\infty$”.
>
> > (Question 2) When a bidder bids independently and its multipliers converge to zero, I would expect a large violation of the RoS constraint, as the bidder tends to win even when the competing bid is larger than their valuation. Why does your result at lines 316-317 not lead the coalition to increase their RoS violation? Is it because thanks to Assumption 3.1, the single bidder of the coalition that actually participate in the auctions has valuation larger than the competing bid in expectation?
>
>
> Yes, the reviewer’s intuition is correct.
>
>
> > (Question 3) Why do the results on the bidder utility (e.g. Theorem 3.1) show that it increases of at least a strictly positive quantity in finite time, while those on the accumulated value show that asymptotically it is no worse than that attained without a coordination mechanism? Is it possible to show that coordination strictly increases the value for sufficiently large T?
>
> In general, it is not possible to show that coordination strictly increases the total value even for sufficiently large T.  In Theorem 4.1 (which doesn’t assume Assumption 3.1), if the outside bid is always sufficiently large, then the coalition never wins under either coordinated or independent bidding, so the total value is always zero in both cases.
>
> In Theorem 4.2 (which assumes Assumption 3.1), the benchmark is over all coordination mechanisms $\mathcal G$. Since our algorithm $A$ itself is a valid coordination mechanism, it is included in this class. Therefore, the guarantee can only be weak (i.e., no worse than any $\mathcal G$), and strict improvement is impossible.

---

> > ### Author Rebuttal · Reviewer_aUCD · 2026-04-04
> >
> > I thank the authors for their detailed response. I still think that Theorem 4.1 is somewhat weak. It would be worthwhile to investigate when the total value under coordinated bidding is weakly larger than that under independent bidding for finite $T$ rather than asymptotically, and to characterize the instances in which it is strictly larger. A similar result for non-i.i.d. case would help strengthen the paper further. I therefore maintain my original score.

---

> > > ### Author Response · Authors · 2026-04-07
> > >
> > > We thank the reviewer for the suggestions. We agree that these directions would enhance the paper. The main challenge for the finite-$T$ comparison is that our current proof is inherently asymptotic: it compares limiting multipliers, but does not imply a round-by-round utility or value gap. Even if coordination has a weakly better limiting multiplier, the per-round utility gap need not be positive, so we cannot simply remove the limit from the argument. Therefore, pursuing these directions would require new techniques beyond our current scope.

---

### Official Review · Reviewer_tELu · 2026-03-21

**Soundness:** 3
**Presentation:** 3
**Significance:** 2
**Originality:** 2
**Overall Recommendation:** 4
**Confidence:** 3

**Summary:**

The paper studies coordination among value-maximizing auto-bidders in repeated second-price auctions, motivated by applications in online advertising. Rather than allowing members of a coalition to bid independently, it proposes a simple coordination strategy in which, at each round, only the coalition member with the highest realized value submits a bid, while all others bid zero. The main contribution is to show that this coordinated strategy can outperform independent bidding. On the theory side, the paper proves that, under a specific distributional assumption, coordination improves each bidder’s utility and thereby reduces RoS constraint violations for any overbidding auto-bidding algorithm. It further shows that, for a class of mirror-descent auto-bidders, coordination asymptotically improves the coalition’s total value as  $T \to \infty$. The paper also extends the analysis to the non-i.i.d. setting, where it still establishes improvements at the coalition level. On the empirical side, the paper shows through experiments that coordinated bidding generally achieves higher total value and lower RoS violation than independent bidding, providing evidence in support of the theoretical results

**Compliance With Llm Reviewing Policy:**

Affirmed.

**Key Questions For Authors:**

- How might the results extend beyond second-price auctions, for example to first-price auctions?

- If the model were generalized without Assumption 3.1, what kind of guarantees or qualitative results should one expect?

**Limitations:**

yes.

**Strengths And Weaknesses:**

Strengths:
- The paper studies a practically relevant problem in online advertising.
- It provides a meaningful theoretical analysis of the proposed coordination rule, including conditions under which coordination improves bidder utility and RoS behavior, as well as coalition-level results for mirror-descent auto-bidders.
- The theoretical claims are complemented by experiments on both synthetic and real-world data.

Weaknesses:
- The main theoretical guarantees rely on Assumption 3.1, and it is not entirely clear to what extent the conclusions extend beyond this setting.
- The analysis is restricted to second-price auctions.
- In the non-i.i.d. setting, the guarantees become weaker and only hold at the coalition level.

---

> ### Author Rebuttal · Authors · 2026-03-30
>
> > (Weakness 1 & Question 2) The main theoretical guarantees rely on Assumption 3.1. If the model were generalized without Assumption 3.1, what kind of guarantees or qualitative results should one expect?
>
> We would like to clarify that Assumption 3.1 is only needed for the per-bidder RoS/utility guarantee (Theorem 3.1), but not for our value-improvement conclusion. In particular, Theorem 4.1 shows that coordinated bidding achieves a weakly higher total coalition value than independent bidding for mirror-descent auto-bidders even without Assumption 3.1. Thus, the qualitative conclusion that coordination improves coalition-level value continues to hold more broadly.
>
> Moreover, Assumption 3.1 is not just an assumption that implies RoS/utility improvement for every bidder.  Our Theorem 3.1 shows that it is also _necessary_: if it fails, then there exist settings where coordination can hurt each bidder’s utility. Therefore, without Assumption 3.1, one should not expect a universal per-bidder RoS improvement, but total value improvements remain robust.
>
> > (Weakness 2 & Question 1) The analysis is restricted to second-price auctions.  How might the results extend beyond second-price auctions, for example to first-price auctions?
>
>
> Before studying coordination, one should understand independent auto-bidding first.  In second-price auctions, shading the bid by a uniform constant is known to be optimal in the independent case, and our analysis for coordination builds on that property.
>
>
> In first-price auctions however, the optimal auto-bidding algorithm even for the independent case is not fully understood yet.  Uniform bid shading is not optimal, and only very recently has some work explored (independent) No-Regret Online Autobidding Algorithms in First-price Auctions (Li et al, NeurIPS 2025), showing that the first-price algorithm can be significantly different and more complicated than the second-price algorithm.  Extending our current analysis to coordinated bidding in first-price auctions thus requires new techniques, which is beyond the current scope.  We expect the high-level effect of reducing within-coalition competition to persist, but the formal guarantees would depend on the underlying bidding dynamics.
>
>
> > (Weakness 3) In the non-i.i.d. setting, the guarantees become weaker and only hold at the coalition level.
>
> In the non-iid setting, the guarantees indeed become weaker and only hold at the coalition level. However, this is unavoidable: in an extreme non-iid scenario where one bidder is strictly dominated by another (e.g., having consistently lower values), an optimal coordinated policy may never select the dominated bidder, so the bidder is worse off under coordination than under independent bidding, making per-bidder improvement impossible. This motivates focusing on coalition-level guarantees. Nevertheless, this limitation can be mitigated in practice by allowing the coordinator to redistribute the total gain among the bidders in the coalition (by monetary transfers) to improve the profit of everyone, as we discussed after Theorem 5.1.

---

> > ### Author Rebuttal · Reviewer_tELu · 2026-04-05
> >
> > Thank you to the authors for their thoughtful rebuttal. After considering their response, I am maintaining my original Weak Accept score.

---

### Decision · Program_Chairs · 2026-04-30

**Decision:**

Accept (regular)

**Comment:**

This papers studies how coalition of bidders can be leveraged to increase the value in repeated second price auctions. Instead of bidding independently, they cooperate so that only the member with highest realized value submits a bid. This is motivated by real-life problem, prominent in online advertising where auctions are key to buy inventory. While the industry mostly uses first-price mechanisms, second-price auction offers a convenient framework that allows neat theoretical analysis, a standpoint leveraged in this work. In this regime, and under some restrictive assumptions, authors shows cooperation to increase each bidder’s utility. The benefit is still preserved - in a weaker sense - when relaxing the assumptions as it applies at the coalition level only. Overall, the paper proposes an interesting theoretical understanding of cooperation in repeated second-price auctions, supported by empirical experiments, which motivates acceptance. This said, reviewers shared concerns about the restrictive nature of the assumption and the limited impact of the result when it relaxes. While this is understandable given the theoretical standpoint of the paper, authors are encouraged to incorporate a discussion on that end, following the rebuttal discussions.